# Understanding the unique S-scheme charge migration in triazine/heptazine crystalline carbon nitride homojunction

Fang Li[1], Xiaoyang Yue[1], Yulong Liao[1], Liang Qiao ®[2] ✉, Kangle Lv ®[3] ✉ & Quanjun Xiang ®[1] ✉

Understanding charge transfer dynamics and carrier separation pathway is challenging due to the lack of appropriate characterization strategies. In this work, a crystalline triazine/heptazine carbon nitride homojunction is selected as a model system to demonstrate the interfacial electron-transfer mechanism. Surface bimetallic cocatalysts are used as sensitive probes during in situ photoemission for tracing the S-scheme transfer of interfacial photogenerated electrons from triazine phase to the heptazine phase. Variation of the sample surface potential under light on/off confirms dynamic S-scheme charge transfer. Further theoretical calculations demonstrate an interesting reversal of interfacial electron-transfer path under light/dark conditions, which also supports the experimental evidence of S-scheme transport. Benefiting from the unique merit of S-scheme electron transfer, homojunction shows significantly enhanced activity for $CO_2$ photoreduction. Our work thus provides a strategy to probe dynamic electron transfer mechanisms and to design delicate material structures towards efficient $CO_2$ photoreduction.

Solar-driven semiconductor photocatalysis, an environmentally benign energy conversion method without external energy intake, has emerged as a promising solution for alleviating energy and environmental crises[1–3]. Generally, semiconductor photocatalytic reaction involves synergistic cooperation of light absorption, carrier separation, and surface redox reaction[4–6], while, the separation of photogenerated carriers is critical for promoting photocatalytic activity since the photoexcited carriers directly participate in the redox reaction[7–9]. Unfortunately, due to strong Coulomb force and dielectric screening in traditional semiconductors, photoexcited electron–hole pairs of single-component photocatalysts tend to recombine in a short time, leading to unsatisfactory photoconversion efficiency[10,11].

On the other hand, photocatalysts consisting of composite semiconductors can induce an internal electric field (e.g., heterostructures or p–n junctions), which affords an additional driving force to separate photoexcited charge carriers and reduce intrinsic recombination[12,13]. For example, composited semiconductor photocatalytic systems, e.g., type-II or S-scheme, are widely exploited in the regulation of charge transfer kinetics[14–16]. Unlike conventional type-II electron-transfer, S-scheme (direct Z-scheme) electron-transfer preserves the higher redox capacity in two-component semiconductors and enables efficient separation of photogenerated charge carriers through the interfacial electric field. Recently, S-scheme transfer has gained wide attention[17,18] and has evolved as an emerging strategy for the design and manipulation of efficient charge separation.

However, the electron-transfer mechanism of S-scheme is confusable with that of type-II[19] thus posing a great challenge to further manipulate the charge transfer dynamics of S-scheme. Although band alignment has long been used as an effective method to differentiate type-II or S-scheme[20], yet one must be very careful to consider the

[1]State Key Laboratory of Electronic Thin Film and Integrated Devices, School of Electronic Science and Engineering, University of Electronic Science and Technology of China, Chengdu 610054, PR China. [2]School of Physics, University of Electronic Science and Technology of China, Chengdu 610054, PR China. [3]Key Laboratory of Resources Conversion and Pollution Control of the State Ethnic Affairs Commission, College of Resources and Environment, South-Central Minzu University, Wuhan 430074, China. ✉e-mail: liang.qiao@uestc.edu.cn; lvkangle@mail.scuec.edu.cn; xiangqj@uestc.edu.cn

actual photoexcitation process by external light illumination. Specifically, the electron-transfer mechanism in photocatalytic experiments mentioned here refers to the dynamic transfer process of hot electrons under external photoexcitation (non-equilibrium condition), which is fundamentally different from the conventional electron transfer at the Fermi level induced by the work function difference between semiconductor components in a dark environment (equilibrium condition). For instance, in "dark-state", type-II and S-scheme are essentially indistinguishable[21].

Therefore, it is highly desirable, as well as challenging, to design advanced strategies to accurately track the transport process of photogenerated electrons and holes for investigating charge transfer dynamics, even manipulating charge carrier separation[22]. To achieve this goal, it is important to identify a suitable photocatalytic material system. In this regard, crystallized carbon nitride homojunction, combining triazine-based crystalline carbon nitride (TCN) and heptazine-based crystalline carbon nitride (HCN), is an ideal candidate photocatalyst owing to high crystallinity with limited defects, good lattice matching at the interface, efficient charge separation[23–26] and excellent photocatalytic performance, thus, it can work as a model system to explore this question. Nevertheless, the electron-transfer mechanism at the interface of crystallized TCN/HCN homojunction remains elusive.

Previous synthesis methods involve simultaneous crystallization of two crystal phases, leading to the formation of intramolecular homojunction in the obtained products, which inevitably limits the dynamic detection of photogenerated electron transfer at the TCN/HCN interface[27,28]. As a result, the exact electron-transfer mechanism remains under debate. For instance, Liu et al.[29] conceived that photogenerated electrons were delivered from HCN to TCN, whereas Zeng et al.[30] held the opposite view. Despite completely different opinions on electron-transfer direction, both groups speculated photogenerated electrons at the TCN/HCN interface following type-II

transfer. However, this conclusion remains controversial due to the scarcity of robust evidence.

To resolve this issue, in this work, we synthesize high-quality crystallized TCN/HCN homojunction (TH1:4) by merging asynchronous crystallization and electrostatic self-assembly strategy and employ state-of-art in-situ photoemission and scanning probe techniques with direct measurement upon light illumination to determine the S-scheme electron migration direction. MnO$_x$ and PtO nanoparticles are loaded at the surface as oxidation and reduction cocatalysts, which work as probes under co-excitation of X-ray and visible light during photoemission to determine the actual band alignment under photoexcitation. The potential difference between TCN and HCN, as well as the change in the surface potential were disclosed intuitively by in situ Kelvin probe force microscopy (KPFM), demonstrating the S-scheme transfer of photogenerated electrons at the TCN/HCN interface. Further theoretical calculations confirm the reversal of the interfacial electron-transfer path in TCN/HCN homojunction following an S-scheme transport mechanism. Taken together, by virtue of the unique advantages of S-scheme electron transfer, the TCN/HCN homojunction demonstrates strikingly enhanced charge separation efficiency and photoreduction CO$_2$ activity.

## Results and discussion
### Material preparation and characterization
As depicted in the transmission electron microscopy (TEM) images (Fig. 1c, f; Supplementary Fig. 1a, b), the synthesized TCN and HCN exhibit nanotube and nanosheet-like morphologies, respectively. The clear lattice fringes in high-resolution TEM (HRTEM) images indicate the high crystallinity of TCN and HCN (Fig. 1a, d), and the obtained interplanar spacings of 0.35 and 0.33 nm correspond to the (002) planes of TCN and HCN, respectively[31–33]. Furthermore, simulated scanning tunneling microscopy (STM) images unveil triazine

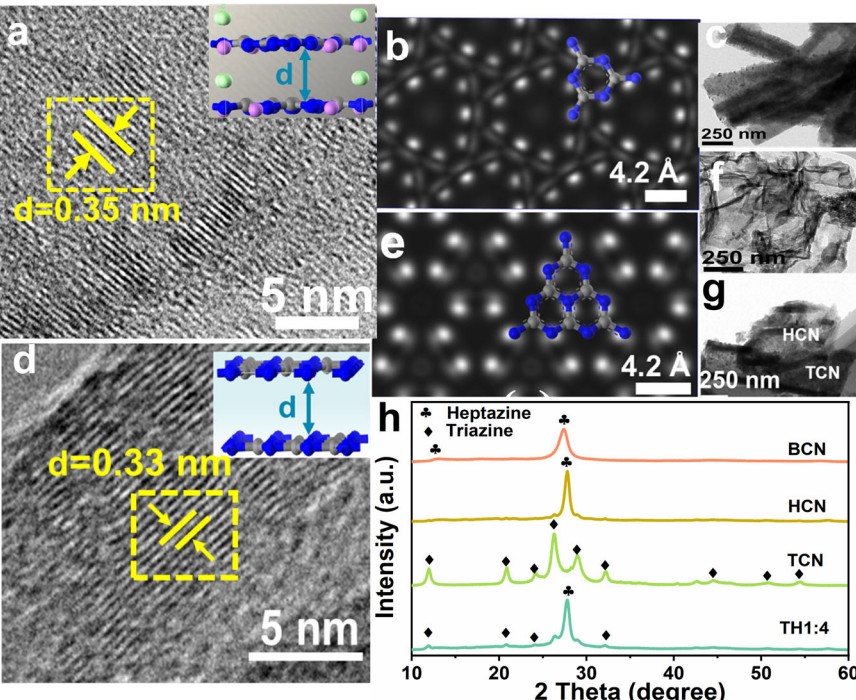

**Fig. 1 | Crystal structure and morphology characterization. a** HRTEM image and corresponding crystal structure of TCN. Color scheme: N, blue; C, gray; Cl⁻, green; and, Li⁺, pink. **b** Simulated scanning tunneling microscopy (STM) image of TCN (The (002) crystal plane is selected for the STM simulation, and the applied bias voltage is −1.8 eV). The inset shows the triazine structural units of TCN. **c** TEM image of TCN. **d** HRTEM image of HCN, the corresponding crystal structure is shown in the inset. **e** Simulated STM image of HCN, the inset displays the heptazine structural units of HCN. **f** TEM images of HCN and **g** TH1:4. **h** XRD patterns of BCN, HCN, TCN, and TH1:4.

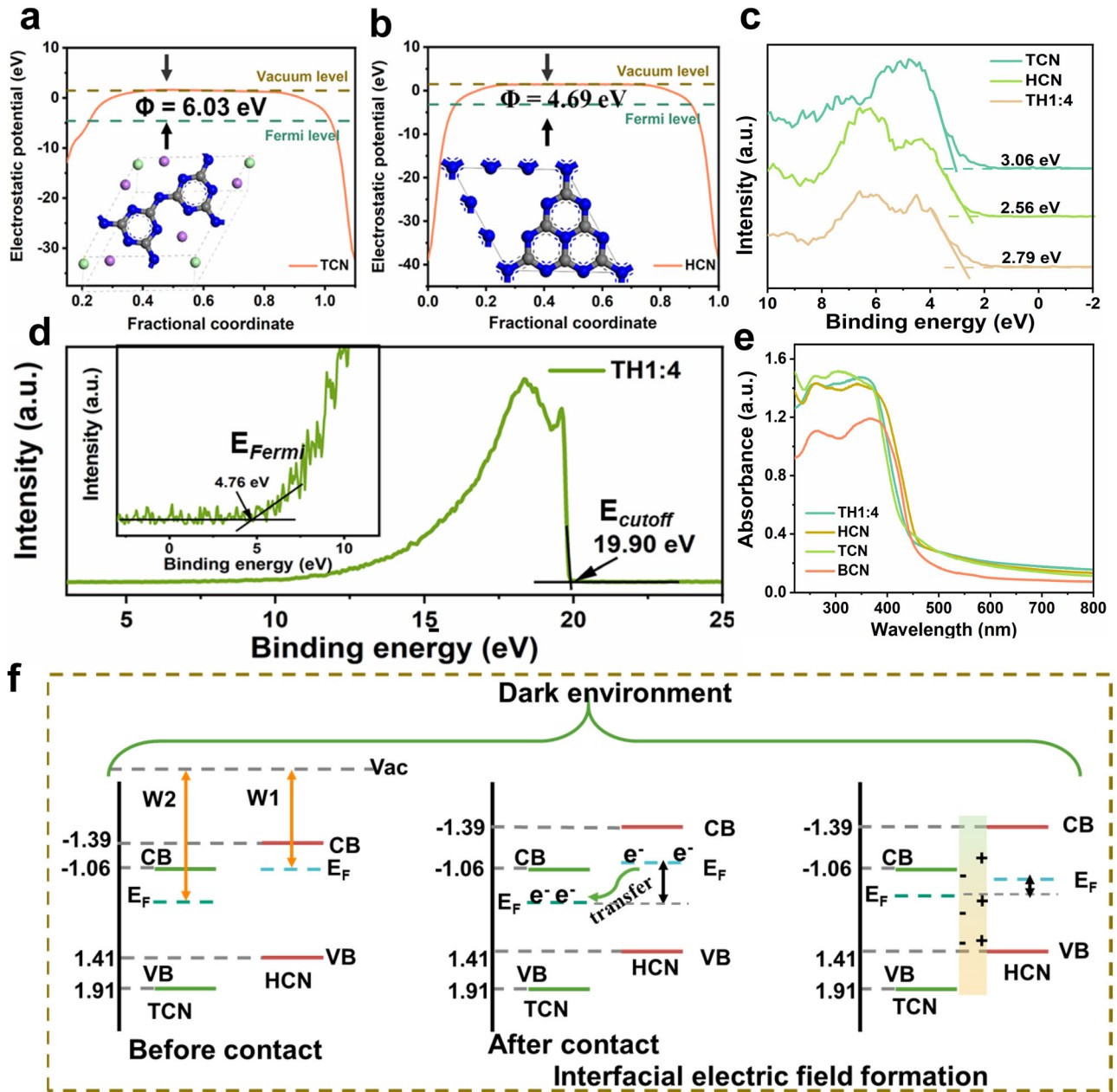

**Fig. 2 | Formation of the interfacial electric field. a** Work functions of TCN and **b** HCN (The insets show the top view of the corresponding slab models). **c** Valence band XPS spectra of TCN, BCN, and TH1:4. **d** UPS spectra of TH1:4. **e** DRS spectra of BCN, HCN, TCN, and TH1:4. **f** Illustration of the formation mechanism for the interfacial electric field.

and heptazine structures as the basic units of TCN and HCN (Fig. 1b, e; Supplementary Fig. 2). After acid treatment, the surface of HCN was positively charged and assembled into a homojunction (TH1:4, i.e., the ratio of triazine/heptazine is 1:4) with negatively charged TCN via electrostatic self-assembly (Supplementary Figs. 3 and 4). As expected, TH1:4 demonstrates interlaced growth of nanosheets and nanotubes (Fig. 1g, Supplementary Figs. 1c and 5), with lattice stripes of two crystalline phases appearing at the interface, indicating the successful formation of homojunction (For detailed information on the atomic arrangement at the interface, please refer to Supplementary Fig. 6). Meanwhile, the X-ray diffraction (XRD) patterns of the synthesized crystallized carbon nitride homojunction also demonstrate two characteristic peaks of triazine and heptazine (Fig. 1h)[34–36], further confirming the successful preparation of TCN/ HCN homojunction.

## Formation of the interfacial electric field

Theoretical calculations are performed to determine the work functions of TCN and HCN, which are 6.03 and 4.69 eV, respectively (Fig. 2a, b, Supplementary Fig. 7, Supplementary Table 1). The difference in work function provides a prerequisite for the formation of an interfacial electric field between TCN and HCN. Furthermore, electronic properties and the band edge positions of the homojunction are characterized by diffuse reflection spectroscopy (DRS) (Fig. 2e, Supplementary Fig. 8), valence band X-ray photoelectron spectroscopy (VB XPS) (Fig. 2c), and ultraviolet photoelectron spectroscopy (UPS) (Fig. 2d)[37]. From DRS measurement along with the Kubelka–Munk analysis, the band gaps of TCN, HCN, and homojunction are determined to be 2.97, 2.80, and 2.92 eV (Fig. 2e, Supplementary Table 2)[38,39], respectively. From XPS and UPS[40–45], the VB positions of TCN, HCN, and homojunction are determined to be 1.91, 1.41, and

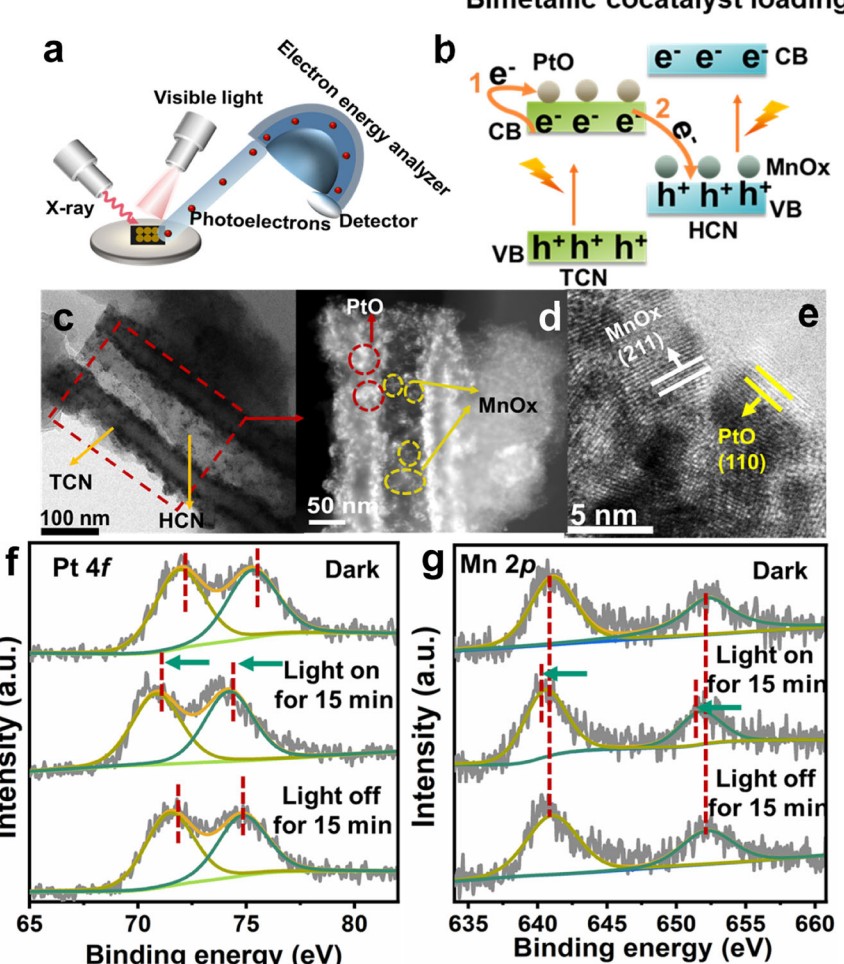

**Fig. 3 | Dynamic tracking S-scheme electron transfer. a** Schematic diagram of the in-situ XPS measurement, the light source used is a xenon lamp. **b** Mechanism diagram of photogenerated electron transfer at the interface of TH1:4 after bimetallic cocatalyst loading. **c** TEM image of TH1:4@PtMn. **d** Scanning TEM (STEM) image of TH1:4@PtMn. **e** HRTEM image of TH1:4@PtMn. **f** XPS Pt 4*f* spectra of TH1:4@PtMn. **g** XPS Mn 2*p* spectra of TH1:4@PtMn.

1.64 eV, respectively. Based on the obtained band edge positions and work function of TCN and HCN, the formation of the built-in electric field at the TCN/HCN interface in the dark environment is illustrated in Fig. 2f. Owing to the smaller work function of HCN in homojunction, electrons will migrate from the Fermi level of HCN to that of TCN, leading to the formation of an internal built-in electric field directed from HCN to TCN.

**Dynamic tracing of S-scheme electron transfer**

Since metallic material can provide a large number of free electrons, thus it can be used as a price indicator to sense the loss and gain of electron transfer. Here, bimetallic cocatalysts, e.g., $MnO_x$ and PtO, were loaded on homojunction as electron transfer indicator, and in-situ XPS with xenon lamp illumination was used to monitor the electron transfer (Fig. 3a, b, Supplementary Fig. 9). Specifically, $MnO_x$ and PtO bimetallic cocatalysts were embedded in both sides of the oxidation photocatalyst (TCN) and the reduction photocatalyst (HCN) by photodeposition, respectively (Supplementary Figs. 10 and 11), and the electron transfer was determined by detecting the change in the binding energy of metal elements before and after light irradiation.

The morphology of TCN/HCN@PtMn is shown in Fig. 3c, d, wherein nanoparticle-like PtO and flake-like $MnO_x$ appear at the interface of nanosheets and nanotubes. As expected, PtO is mainly concentrated on the surface of TCN, while $MnO_x$ is mainly on the surface of HCN, as confirmed by the energy dispersive spectrometry

(EDS) spectra (Supplementary Fig. 12). HRTEM images show that the lattice spacing of 0.213, and 0.241 nm are attributed to the (111) plane of PtO and the (211) plane of $MnO_2$ (Fig. 3e), indicative of the successful loading for bimetallic cocatalysts. Furthermore, the binding energies of Pt $4f_{5/2}$ (75.3 eV), Pt $4f_{7/2}$ (72.0 eV), Mn $2p_{1/2}$ (652.3 eV), and Mn $2p_{3/2}$ (641.1 eV) indicate that the valence state of Pt is +2, similar in PtO[46,47] and the valence state of Mn is between +3 and +4[48].

Interestingly, compared with that of the dark state, the binding energy of Mn 2*p* shifted to the left by 0.6 eV after 15 min of illumination and returned to the "dark-state" energy position after light-off (Fig. 3g). The decrease in the binding energy of Mn 2*p* upon light illumination indicates the gain of electrons for $MnO_x$. In other words, the photogenerated electrons on the conduction band of TCN are transferred to the Mn 2*p* orbital on HCN by the attraction of an electric field force.

Meanwhile, the change of binding energy of Pt 4*f* on TCN should be opposite to that of Mn 2*p*. But this is not the case, we noticed that the binding energy of Pt 4*f* under light illumination exhibited a similar leftward shift of 1.1 eV compared to that of the dark state and did not completely return to the "dark-state" energy position after light-off (Fig. 3f). This involves two electron transport processes of PtO loaded on TCN. As shown in the schematic diagram (Fig. 3b), the first process is direct photoexcited electron transfer from TCN to PtO (i.e., leading to decreased binding energy for Pt 4*f*), and the second process is indirect electron transfer from PtO to $MnO_x$ anchored on HCN driven by the interfacial electric field (i.e., leading to increased binding energy

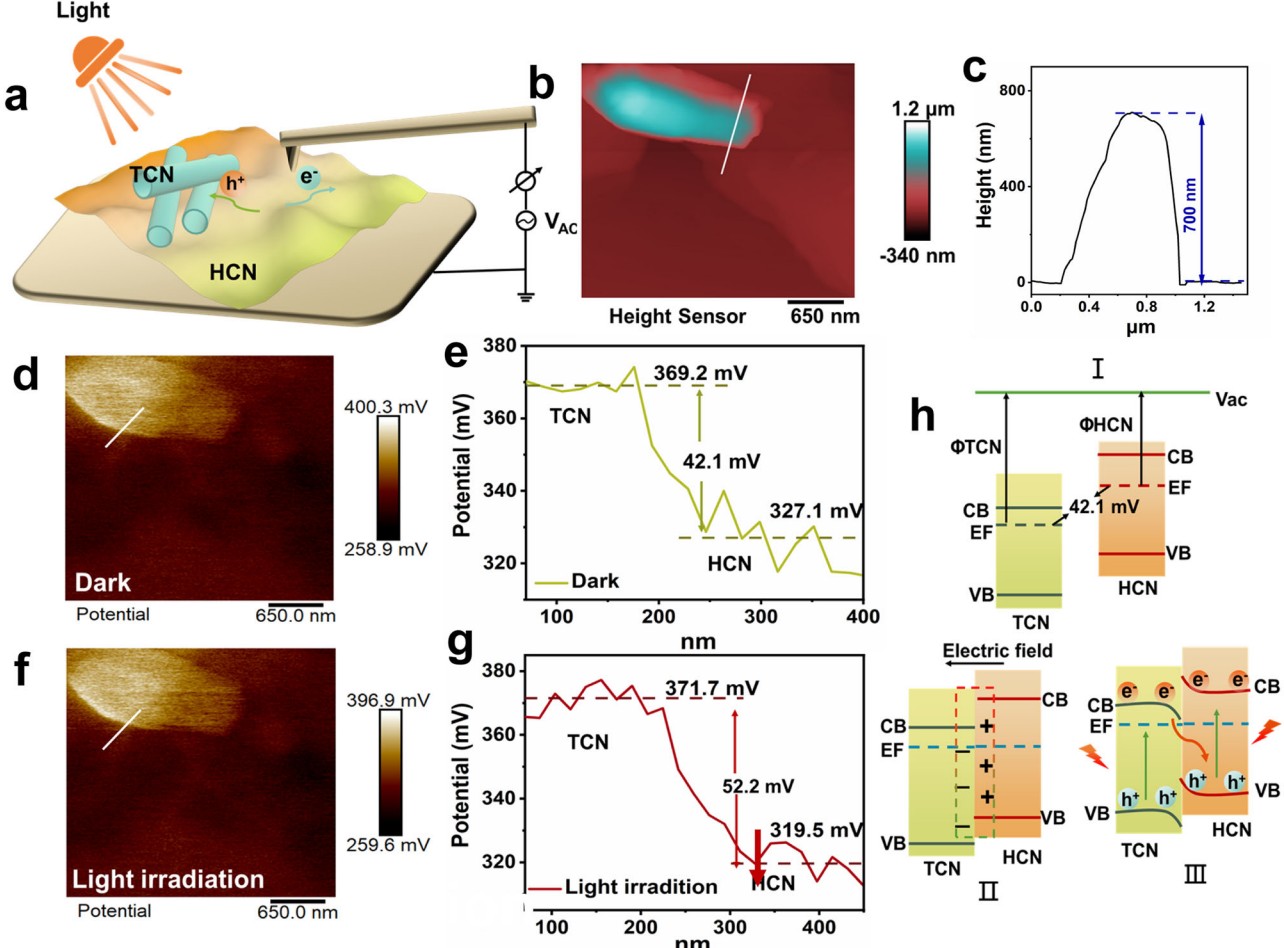

**Fig. 4 | Verification and mechanism reveal of S-scheme electron migration.**
**a** Illustration of light-assisted KPFM measurement, the light source used is a xenon lamp, **b** AFM image of TH1:4, and **c** corresponding height distribution curve. **d**, **e** The distribution of potential difference at the interface of TH1:4 in a dark environment. **f**, **g** The distribution of potential difference at the interface of TH1:4 under illumination. **h** Schematic of the S-scheme migration of photogenerated carriers induced by the electric field at the TH1:4 interface. Processes I and II represent the formation of the built-in electric field owing to the difference in the work function, and process III represents the separation of the charges following an S-scheme mechanism induced by the built-in electric field.

for Pt 4$f$). In an ideal case, if the two processes are balanced, then there should be no binding energy shift for the Pt 4$f$ core-level peak. Apparently, the slight change of binding energy of Pt 4$f$ before and after illumination reflects that the first electron transport process is the dominating procedure (Supplementary Fig. 13). Although the electron transfer path involved in PtO is relatively complicated, the electron transfer on $MnO_x$ provides evidence that the photogenerated electrons are transferred from TCN to HCN.

To further prove that photogenerated electrons at the interface of TCN/HCN homojunction follow S-scheme transfer, light-assisted KPFM was employed to probe the change in the local potential of the homojunction before and after illumination (Fig. 4a)[49–51]. Atomic force microscope (AFM) images and corresponding height distribution curves indicate that the selected region is the interface between TCN and HCN in the homojunction (Fig. 4b, c). As shown in Fig. 4d, e, the distribution of two-phase surface potential in the "dark-state" shows that the surface potential of TCN in the homojunction is larger than that of HCN, creating a gap of 42.1 mV between them. According to the equation: $V_{CPD} = -(\Phi_{tip} - \Phi_{sample})/e$ ($V_{CPD}$ means the contact potential difference between the tip and the sample surface; $\Phi_{tip}$ and $\Phi_{sample}$ represent the work function of tip and sample)[52,53], the work function of TCN in the homojunction is larger than that of HCN, which is consistent with the theoretical calculation results. After 15 min of light illumination, the surface potential at the conjunction of TCN and HCN

in the same region changed obviously. Compared with that of the "dark-state", the surface potential of TCN increases from 369.2 to 371.7 mV (Fig. 4d, e), while that of HCN decreases from 327.1 to 319.5 mV (Fig. 4f, g). Changes in the KPFM surface potential of TCN/HCN homojunction induced by light irradiation directly visualize the dynamic transfer of photogenerated electrons from TCN to HCN in both real-space and real-time scales (Fig. 4h), thus confirming the S-scheme charge transfer mechanism.

**Theoretical simulation concerning the inversion of the electron transport path**
Theoretical calculations were used to further understand the mechanism of photogenerated electron transfer between TCN and HCN (Supplementary Fig. 14, Supplementary Table 3). The work function of the TCN/HCN homojunction is 5.86 eV (Fig. 5a), accompanied by an interlayer distance of 3.3 Å. Meanwhile, the electrostatic potential of HCN is deeper than that of TCN, indicating that electrons tend to transfer from HCN to TCN[54]. Note that conventional theoretical calculation can only simulate the static charge distribution on the semiconductors, mimicking the case of a "dark-state" environment, thus it cannot be directly used to analyze the mechanism of photogenerated carrier migration during light excitation. Here to simulate the light excitation condition, we introduce an electrical field during calculation (details can be found in the theoretical description).

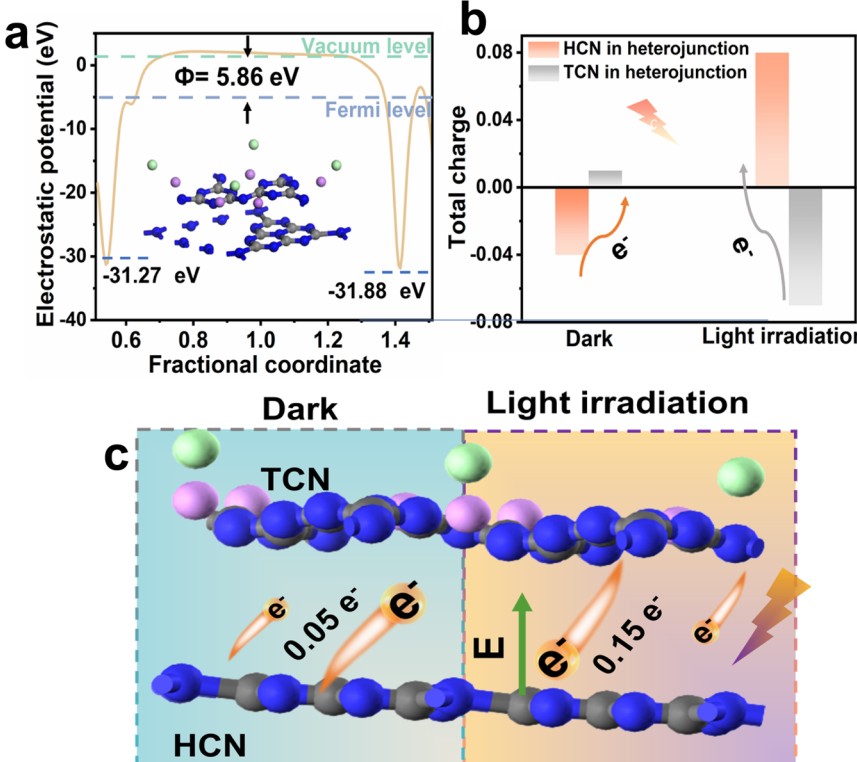

**Fig. 5 | Theoretical insight into electrical changes caused by S-scheme electron transfer. a** The work function of the crystallized carbon nitride homojunction, where the inset shows the optimized model of the homojunction. **b** The Mulliken charge analysis of the crystallized carbon nitride homojunction. **c** Schematic of the interlayer charge transfer for crystallized carbon nitride homojunction under dark and illumination.

Excitingly, Mulliken charge analysis reveals opposite electron transfer pathways of the TCN/HCN homojunction under dark and illumination environments (Fig. 5b). Under dark conditions, electron depletion occurs on the surface of HCN, whereas electron enrichment emerges on the surface of TCN, revealing that electron transfers from HCN to TCN. However, under light conditions, the electron-transfer path is completely reversed (Fig. 5c). This is also confirmed by 2D charge density difference maps (Supplementary Fig. 15) for the TCN/HCN homojunction, further supporting the photogenerated electrons at TCN/HCN interface follow the S-scheme transfer mechanism.

**Impact on photocatalytic activity**

To investigate the effect of S-scheme electron transfer on photocatalytic activity of TCN/HCN homojunction, designed $CO_2$ photoreduction experiments were carried out (Supplementary Fig. 16). Among the as-prepared samples, TCN/HCN homojunction with optimized ratio (TH1:4) demonstrates the best activity for $CO_2$ photoreduction in the absence of sacrificial agents and cocatalysts, i.e., CO and $CH_4$ yields exceed that of bulk carbon nitride (BCN) by 2.09 and 9.45 times, respectively (Fig. 6a, b). In addition, the TH1:4 sample possesses the highest electron consumption rate in the photoreduction of $CO_2$ (Fig. 6c)[3]. Furthermore, the effect of different ratios of triazine/heptazine on $CO_2$ photoreduction activity for TCN/HCN homojunction was also evaluated (Supplementary Figs. 17 and 18, Supplementary Tables 4 and 5). Generally, with the increase of the heptazine phase concentration in the homojunction, the $CO_2$ photoreduction activity first gradually increases and then starts to decay after reaching a threshold value (Fig. 6d, e).

To verify whether the crystal phase ratio can impact the electron transfer mechanism, i.e., from S-scheme to type-II, the dynamic electron transfer of a TH1:1 sample supported with bimetallic cocatalyst was also investigated by in-situ XPS (Supplementary Figs. 19

and 20). The results demonstrate that the electrons in TH1:1 still follow the S-scheme migration, suggesting the triazine/heptazine phase ratio does not change the electron migration behavior. Therefore, the observed changes in activity induced by the triazine/heptazine phase ratio can be ascribed to the variations in the built-in electric field strength. When the triazine/heptazine phase ratio reaches 1:4, the interface electric field strength is the strongest, thus showing the highest photocatalytic activity (Fig. 6d, e, Supplementary Fig. 21).

Further in-situ diffuse reflectance infrared Fourier-transform spectroscopy (DRIFTS) on the optimized TH1:4 sample identifies $CH_3^-$ (1456 cm$^{-1}$)[55], b-$CO_3^{2-}$ (1506 cm$^{-1}$)[56], $HCOO^-$ (1539 cm$^{-1}$)[57], and $CO_3^{2-}$ (1557 cm$^{-1}$)[58] as intermediates in the photoreduction of $CO_2$ (Supplementary Fig. 22, Supplementary Fig. 23). The $^{13}C$ isotope tracking experiment shows that the $CH_4$ and CO generated during the photoreduction experiments come from the $^{13}CO_2$ reaction gas (Fig. 6f)[59], and the $O_2$ molecular is also detected (Supplementary Fig. 24). Electrochemical measurements were also used to analyze the mechanism of photocatalytic activity enhancement for TH1:4 sample. Among all the prepared samples, the optimized sample TH1:4 possesses the smallest arc radius in the electrochemical impedance spectroscopy, implying improved charge separation ability (Supplementary Fig. 25)[60,61]. The similar overpotentials of the TH1:4 and HCN in the linear sweep voltammetry (LSV) measurement indicate that the S-scheme TCN/HCN homojunction retains a strong reducing ability (Supplementary Fig. 26).

Taken together, we have experimentally verified that S-scheme electron transfer occurs in TCN/HCN homojunction. Our work also highlights that modulating electrons at the homojunction/heterojunction interface and enabling the S-scheme transfer mechanism can be potentially a strategy to facilitate charge separation and promote photocatalytic activity.

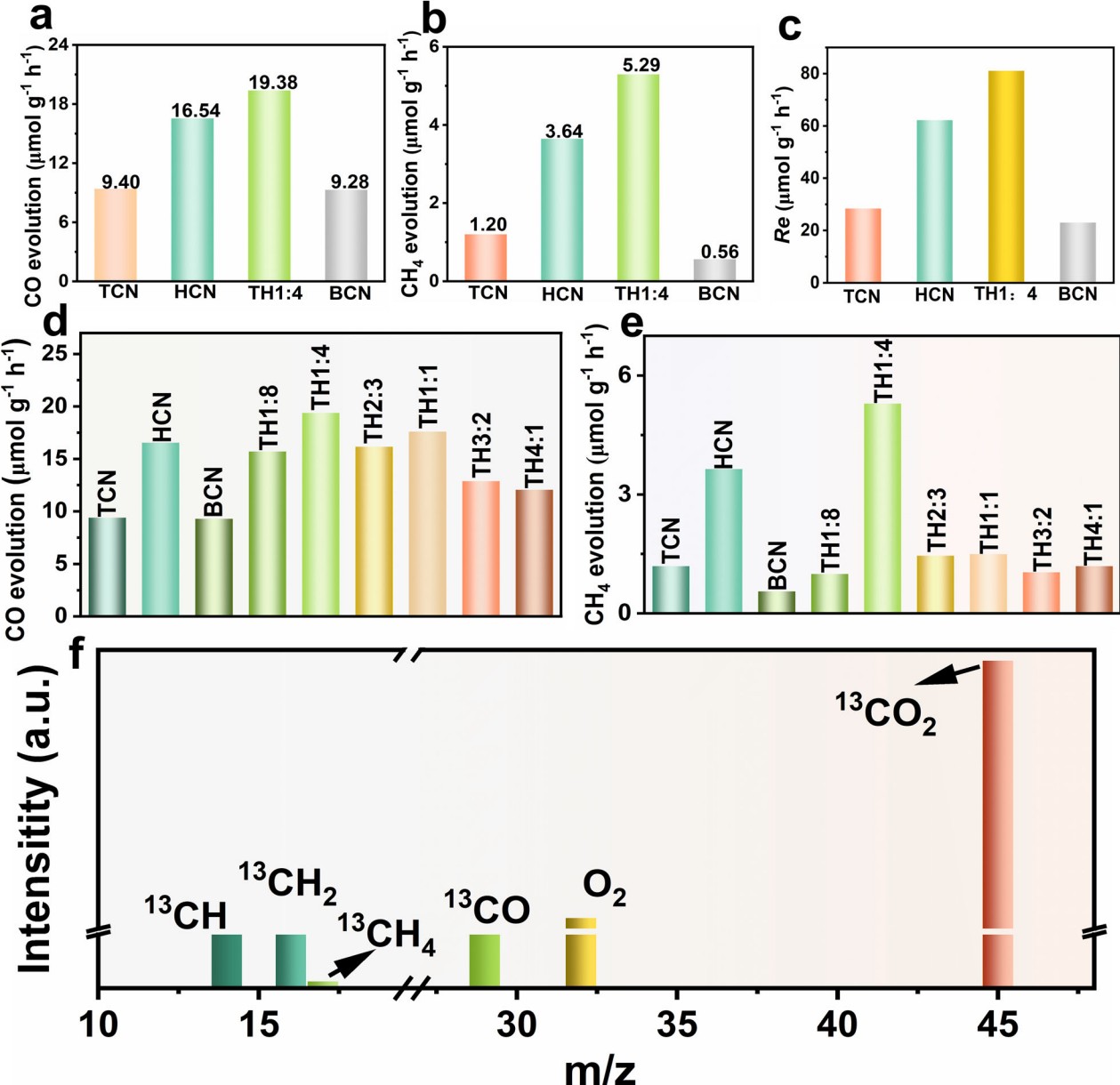

**Fig. 6 | Effect of S-scheme electron transfer on photoreduction CO₂. a** Yields of CO and **b** CH₄ of TCN, HCN, TH1:4, and BCN samples. **c** The electron consumption rate of the as-prepared samples during the photoreduction of CO₂, **d** Yields of CO, and **e** CH₄ over homojunction formed by different ratios of TCN and HCN. **f** [13]C isotope tracing experiment during the photoreduction of [13]CO₂ among TH1:4.

In summary, the crystallized carbon nitride homojunction (TH1:4) was prepared to explore the interfacial electron-transfer mechanism. A combination of theoretical simulations and experimental characterization demonstrated that photogenerated electrons at the crystallized carbon nitride interface followed S-scheme transfer from TCN to HCN. Specifically, In situ, XPS monitored the S-scheme transfer of interfacial photoelectrons from the triazine phase to the heptazine phase using a bimetallic co-catalyst loaded on TH1:4 as a probe. Furthermore, in situ KPFM traced the dynamic S-scheme transfer of interfacial photogenerated electrons in real time based on the intuitively recorded changes in the surface potential of TH1:4. Theoretical calculations revealed that the electron transfer path at the interface of the crystalline carbon nitride homojunction was reversed by the interfacial electric field upon light irradiation, enabling electrons to follow S-scheme transport. Benefiting from the unique advantages of S-scheme electron transfer, TH1:4 exhibited significantly enhanced charge separation efficiency and photoreduction CO₂ activity. This work paves a potential way for identifying the dynamic transfer mechanism of photogenerated carriers from the atomic level, and even for designing and tuning the transfer of photogenerated electrons in an advanced manner.

## Methods

### Materials

Potassium chloride (KCl, 99.5%), lithium chloride (LiCl, 98.5%), and methanol (CH₄O) were provided by Chengdu Chron Chemicals Co., Ltd. Manganese sulfate (MnSO₄) was supplied by Shanghai Macklin Bio-Chem Technology Co., Ltd. Sodium iodate (NaIO₃, 99%), melamine (C₃H₆N₆, 99%) chloroplatinic acid hexahydrate (H₂PtCl₆·6H₂O, 99.95%) were obtained from Shanghai Aladdin Bio-Chem Technology Co., Ltd.

### Synthesis of heptazine crystallized carbon nitride (HCN)

A 10 g of melamine was calcined at 500 °C for 4 h. Subsequently, 600 mg of the obtained pale yellow product was ground in a crucible with 3.3 g KCl and 2.7 g LiCl, the ground mixture was heated to 550 °C in a muffle furnace, and calcined for 4 h under a nitrogen atmosphere. Ultimately, the obtained product was washed with boiling water and collected by centrifugation. The resulting product was named HCN, which has a yield of roughly 50–60%.

### Synthesis of triazine-based crystallized carbon nitride (TCN)

Totally, 600 mg of melamine, 3.3 g of KCl, and 2.7 g of LiCl were ground in a crucible, and then the mixture was placed in the crucible and calcined in a nitrogen atmosphere at 550 °C for 4 h. The obtained product was washed with boiling water and collected by centrifugation. The resulting product was labeled as TCN with a yield of approximately 50–60%.

### Synthesis of bulk carbon nitride (BCN)

A 10 g of melamine was calcined at 550 °C for 4 h. The obtained product was BCN. The resulting product was labeled as TCN with a yield of approximately 50%.

### Synthesis of acid-treated heptazine-based crystallized carbon nitride (HHCN)

Totally, 450 mg of HCN was added to 200 mL of the hydrochloric acid solution for ultrasonic treatment for 1 h. Afterward, the mixture was stirred for 4 h. The final product was collected by centrifugation and dryness. The resulting product was marked as HHCN with a yield of approximately 50–60%.

### Synthesis of triazine and heptazine-based crystallized carbon nitride homojunction in different proportions

Taking the synthesis of a specific ratio of crystallized carbon nitride homojunction (TH1:4) as an example, TH1:4 represents the mass ratio of TCN and HHCN. 120 mg of HHCN was dispersed in 100 mL of deionized water with 30 mg of TCN added. The mixture was sonicated for 30 min and stirred for 4 h. Then the desired product was obtained by centrifugation and dryness. The resulting product was marked as TH1:4 with a yield of approximately 50–60%.

### Synthesis of triazine-based crystalline carbon nitride-supported PtO (TCN@Pt)

Totally, 100 mg of TCN was dispersed in 200 mL of the aqueous solution containing 10% methanol. Then, 13.27 mL of chloroplatinic acid solution (1 mg/mL) was added, and the Pt species were deposited on TCN by photoreduction. The loading amount of Pt was 5%. The resulting product was marked as TCN@Pt with a yield of approximately 50–60%.

### Synthesis of heptazine-based crystalline carbon nitride-supported $MnO_x$ (HHCN@Mn)

Totally, 100 mg of HHCN was dispersed in 200 mL of deionized water, using $NaIO_3$ as an electron scavenger. Mn species originating from manganese sulfate (13.74 mg) were deposited on HHCN by photo-oxidation. The loading amount of Mn was 5%.

### Synthesis of crystallized carbon nitride homojunction supported with bimetallic cocatalysts (TH1:4@PtMn)

TCN@Pt and HHCN@Mn were mixed and stirred according to the mass ratio of 1:4, and the specific synthesis steps were the same as those of TH1:4. The resulting product was marked as HHCN@Mn with a yield of approximately 50–60%.

### Sample characterization

XRD patterns were supplied by Bruker D8 Advance. TEM image and EDS images were captured by Tecnai G2 F20 with EDS Inca X–Max (Oxford). Scanning electron microscopy (SEM) images were given by Gemini SEM 300. In-situ XPS measurement was operated on a Kratos-Axis Ultra DLD instrument using a xenon lamp as the light source. The binding energy of C1s at 284.8 eV was used for energy calibration. VB XPS spectra and UPS of the samples were collected by Escalab Xi+ (Thermo Scientific), and UV–vis DRS was determined by the UV-2600 system (Shimadzu). The content of K and Li was obtained from ICP-OES (PE Avio 200) and the content of H was obtained from O, N, and H co-meter (LECO ONH836). AFM images and KPFM images were captured by Bruker MultiMode 8, and the probe used was a silicon probe. During the measurement, a xenon lamp was used as the light source, and the light irradiation time was 15 min. In this case, the powder sample was dispersed in an ethanol solution and then dropped onto the silicon wafer to form a uniform film for testing.

In-situ DRIFTS characterization was operated on a Nicolet iS50 FTIR spectrometer. $^{13}C$ isotope tracing experiment was performed by mass spectrometry (Finnigan MAT 271). The $O_2$ generated during the $CO_2$ photoreduction process was detected by a gas chromatograph (GC14C, Shimadzu) with a TCD detector.

### Theoretical details

The theoretical calculation part was completed by the CASTEP code using ultrasoft pseudopotential. The geometric optimization process used the Perdew–Burke–Ernzerh equation originating from generalized gradient approximation (GGA)[62]. The convergence criterion for energy and force are $1.0*e^{-5}$ eV/atom and 0.03 eV/A. The Grimme method was adopted to describe the van der Waals interaction[63]. Cutoff energy and the value of K-point mesh were determined to be 500 eV, and 3*3*1 after convergence tests. The slab models of TCN and HCN were obtained by cleaving (002) crystal planes of bulk TCN and bulk HCN. The monolayer slab model of TCN and the monolayer slab model of HCN are geometrically optimized accompanied by a vacuum layer thickness of 15 Å. The information on the work function is then obtained by property calculations. The crystallized carbon nitride homojunction model was constructed using a monolayer slab of TCN and a monolayer slab of HCN crystalline surfaces and the lattice mismatch was within 5%. The vacuum layer thickness of the homojunction model is set to 20 Å. The optimized structural parameters for the homojunction model are as follows, $a = b = 8.0234$ Å, $c = 25.1154$ Å. Light irradiation conditions were simulated by an applied electric field in the z-direction.

### Photoreduction of $CO_2$ and photoelectrochemical characterization

The photoreduction experiment of $CO_2$ was completed by an online photocatalytic system (Labsolar-6A). A 30 mg sample was dispersed in 3 mL of ethanol, after 5 min of sonication, and then poured into a flat-bottomed crucible lid to dry. Then it was placed into the quartz reactor with 0.5 mL of deionized water added. Before illumination, the system was evacuated and the $CO_2$ reaction gas was introduced. CO and $CH_4$ produced by photoreduction were detected by the flame ionization detector of the gas chromatography (FULI 979011). Photoelectrochemical characterization was done by a three-electrode system (CHI660E) consisting of a counter electrode (Pt wire), a reference electrode (Ag/AgCl) electrode, and a working electrode. 30 mg of the sample and 30 mg of polyethylene glycol were mixed with ethanol and ground into a slurry, and then the slurry was coated on the FTO conductive glass, covering an area of 1 $cm^2$. After drying at 100 °C for 2 h, the preparation of the working electrode was completed.

### Data availability

The data generated in this study are provided within the manuscript and Supplementary Information file. Source data are provided in this paper.

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

## Acknowledgements
This work was partially supported by the National Natural Science Foundation of China under grant No. 22272019 and 51672312, the Sichuan Science and Technology Program under No. 2022JDRC0096 and 2022ZYD0039, the Open Foundation of State Key Laboratory of Electronic Thin Films and Integrated Devices under grant No. KFJJ202105.

## Author contributions
F.L. conceived the idea, designed the experiments, and wrote the original draft. X.Y. and Y.L. participated in the design and layout of the pictures and reviewed the paper. L.Q., K.L., and Q.X. supervised the work and reviewed and revised the paper.

## Competing interests
The authors declare no competing interests.
