## [Peer Review File · Nature Communications]

Understanding the unique S-scheme charge migration in triazine/heptazine crystalline carbon nitride homojunctionREVIEWER COMMENTS

Reviewer #1 (Remarks to the Author):

This manuscript demonstrates the so-called homojunction between two allotropes of carbon nitride and investigated the charge transfer behavior between the two components. The study of the charge carrier transfer at the interface is of significant importance to deeply understanding the photocatalytic mechanism. The main assumption of this submission is attractive, but more solid characterizations are still lacked.

1. The author assume the formation of homojunction between TCN and HCN. However, there are no such characterizations (such as TEM) to provide direct evidence. More detail information at the interface is highly desired to support the main assumption of the manuscript. It is widely considered that the weak electrostatic interaction between TCN and HCN only generate nanocomposites rather than junctions.
2. The author demonstrates high crystallinity of TCN and HCN. But only lattice fringes that are related to the interplanar spacings were recorded by TEM. The lattice fringes related to the in-plane packing modes (100) are instead solid evidence to prove the high crystallinity of the carbon nitride materials. Also, the absence of diffraction peak at 8 θ of HCN in Figure 1h illustrates that HCN is disorder in the in-plane packing modes of the heptazine units.
3. The chemical compositions of so-called TCN and HCN are still not very clear. Are there any residues of the metal salts? Previous studies by Lotsch and Antonietti claimed that Li⁺ and Cl⁻ ions are presented in PTI/Li⁺Cl⁻ (here TCN) and K⁺ always incorporated in K-PHI (here HCN). Please clarify the difference between TCN, HCN and PTI/Li⁺Cl⁻ K-PHI.
4. Connected the above question, the acid treatment of HCN would lead to ion exchange (K⁺ was replaced by proton). Elemental analysis of the samples before and after acid treatment are involved.
5. Here they author show the nanosheet structure of HCN, which is quite different from the nano rode/fiber morphology of the K-PHI in the previous literature. Please add more explanation about the morphology.
6. In-situ photo-deposition of Pt nanoparticles from chloroplatinic acid are usually observed in literatures, but here PtO rather than Pt is obtained, please comment on this issue.
7. The characterizations by KPFM is not clearly discussed. Did the author conduct the KPFM direct with the powders of HCN and TCN? As I know, it is usually performing this characterization by using the homogeneous film with flat surface.
8. In Figure S15, the CO₂ reduction activity dramatically decreased only after few hours' reaction. Is this derived from the weak interaction between the two components?
9. The authors propose that the electron transfer path of TH1:4 under light irradiation is from the conduction band of TCN to the valence band of HCN. And the electrons in TH1:4@PtMn take another path, i.e., from the conduction band of TCN to PtO, then to MnOx and finally to the valence band of HCN. Does this mean that PtO on TCN and MnOx on HCN also form good contacts and facilitate electron transfer? The photocatalytic activity of TH1:4@PtMn is worth investigating.
10. The XPS Pt 4f and Mn 2p spectra (in dark, light on and light off) of TCN@Pt, HHCN@Mn,

TH1:4@MnPt (MnOx on TCN and PtO on HCN) is beneficial for further understanding of the electron transfer mechanism. Can the author provide these data?

Reviewer #2 (Remarks to the Author):

Key results: Please summarise what you consider to be the outstanding features of the work.
Understanding how the e- and h+, produced in the bulk of a photocatalyst transfer and eventually react with adsorbed substrates, is very important to rationally design and optimize the photocatalyst. The authors aim at clarifying the mechanism of electron transfer at the homojunction of crystalline TCN (triazine carbon nitride) and HCN (heptazine carbon nitride) with a combined experimental (TEM, HRTEM, XRD, DRS, VB XPS, UPS, EDS, KPFM, AFM) and computational approach (DFT calculations). Surface bimetallic cocatalysts are used as probes to verify the scheme e- transfer from TCN to HCN. The investigation has a more general goal to define a strategy to probe the electron transfer mechanism, which is relevant to a broad audience.

Validity: Does the manuscript have flaws which should prohibit its publication? If so, please provide details.

A good part of the presented data is solid, and the investigation was well organised. However, in the present form, the reliability of the computational data is not clear, my concerns are reported in the section "Data & methodology" below. The validity of the computed data should be better discussed and possibly verified with further calculations, as suggested below.

Originality and significance: If the conclusions are not original, please provide relevant references. On a more subjective note, do you feel that the results presented are of immediate interest to many people in your own discipline, and/or to people from several disciplines?
The results are potentially interesting and relevant to a broad audience of people working with photosensythesed systems.

Data & methodology: Please comment on the validity of the approach, quality of the data and quality of presentation. Please note that we expect our reviewers to review all data, including any extended data and supplementary information. Is the reporting of data and methodology sufficiently detailed and transparent to enable reproducing the results?

The methodology used in the DFT calculations is not clear, and therefore the criteria of reproducibility is not met.

1) It is not completely clear what systems were simulated (see below). How many atoms were included in the periodic cells? how many ions?

A clear indication of the modeled system(s) comprising also a proper representation of the system simulated is fundamental to enable reproducing the results. You can find many examples of clear

representation of the periodic simulation box in the literature, as examples: Phys. Chem. Chem. Phys., 2016, 18, 31175–31183, Fig. 1 for representation of the simulation cell for the junction; J. Phys. Chem. C 2018, 122, 7712–7719, Fig. 1 for the representation of g-C₃N₄ supercell model; nature communications 2020 11:4613, Fig. 3, for box representation.

It is not clear how many systems were modeled: TCN alone, HCN alone and the TCN/HCN homojunction? Or only the latter? Are Li⁺ and Cl⁻ included in the simulation box? Please state clearly this information in the method section (a Table might be useful) and represent all of the model systems clearly (see examples above) either in the main text or in the supporting.

2) the separation between the periodic structures along the z-axis is set to 15 Å. This value is smaller than that of 20 Å used in other investigations to avoid spurious interactions between periodic image (see e.g. NATURE COMMUNICATIONS | (2020) 11:4613; J. Phys. Chem. C 2018, 122, 7712–7719; Applied Catalysis B: Environmental 2020, 268, 118381). While the value of 15 Å has been used also in other calculations, my concern here is due to the presence of ions, which could make the interaction between layers be stronger (i.e. longer ranged) than for other systems where neutral layers are modeled. I suggest the author to check whether the separation may affect the results and the conclusions, or if there are other evidences that indicate that the use of this separation does not alter the results on similar systems.

3) What do the authors mean with “simulated scanning tunneling microscopy (STM) images” (lines 125-127, 573, 577) and what they are used for. Do the authors refer to a STM images calculated as in reference: Scientific Data volume 8, Article number: 57 (2021)? If so, I have the following concern. If I understand correctly, in the present manuscript no STM experiment is performed. Could the author comment why the simulated STM images are useful in the present investigation if there is no experimental reference? It is not obvious how a simulated STM image can “unveil triazine and heptazine structures as the basic units of TCN and HCN”: if the structures were used as input for the simulations of the STM spectra, what else could be expected?

4) Please specify how the work functions were calculated.

5) Charge density maps: the analysis of this property has an important role in this investigation to supports the proposed S-scheme mechanism. However, the data presented are not clear, and also the discussion and conclusions reached with the presented data are not convincing in the present form. In particular, in Sup. Inf. Fig. 9: the pictures (a) and (b) are taken from different points of view and are difficult to compare. Exactly the same orientation should be used to allow the reader to make the comparison and verify the conclusion of the authors. Fig (c) and (d): there are literally infinite possible planes on which the charge density can be represented. Depending on which plane is taken completely opposite conclusion can be (in principle) drawn. Why the authors draw their conclusions only one particular plane? How the conclusions drawn on a particular plane can be considered general?

6) Mulliken charges are known to have several drawbacks, and other schemes for calculating the charges partitioning among atoms have been developed. Considering that the authors are proposing a strategy of general interest, it could be useful if they can comment on the validity of this aspect of the computational approach they are proposing.

7) Figures

Some Figures are not clear, and improving their quality would be helpful for the reader:

- Fig 1: the represented molecular structures in insets of subfigure (a) and (d) are not clear. I suggest

using more conventional representation in ball and stick. Also, the atom scheme coloring is a bit confusing, it would be easier for the reader to always have gray color for C atoms and blue for Nitrogen, as used in other Figures, such as supporting Fig 9. Same problem in Fig. 5c, it is very difficult to visualize the structure with the representation used here for C and N, and it is not clear why some yellow balls (nitrogen) are not linked to any carbon atoms.. Also, I find confusing the use of two different molecular representations in the same figure (in fig 5a and b is ball and stick with different colors from the vdw representation in fig 5c).

- Fig 2f: it would be easier to analyse if the y axis values would be aligned among the three subfigures. The quality of the figure is low, both in term of resolution and in terms of “collaging” quality (alignment and styles of the graphics).

- Fig 3e: the writing over the figure is very difficult to read.

8) The description of the Figures is in some case not clear:

- Fig. 4h, contains three subfigures, I, II, III. The legend does not clarify what each of them represent.

- Figs 2b and 5a: what the insets are representing? Why some of the carbon atoms are not represented? Or are they lacking from the model?

- Fig. 2a and 5c: what the pink and cyan balls are representing?

9) Other comments on the text:

- Line 129: “TH1:4, i.e. the ratio of heptazine/triazine is 1:4” is this phrase correct? I find confusing that the order of the initials in the acronym is opposite to the ratio of the components to which the initials refers. It would be clearer as HT1:4 if the ratio H/T is 1:4.

- Line 223-224: here the authors comment on the electrostatic potential, but I cannot find where the results on this property are reported.

Appropriate use of statistics and treatment of uncertainties: All error bars should be defined in the corresponding figure legends; please comment if that’s not the case. Please include in your report a specific comment on the appropriateness of any statistical tests, and the accuracy of the description of any error bars and probability values.

No statistical analysis is reported, the possible errors are not commented.

Conclusions: Do you find that the conclusions and data interpretation are robust, valid and reliable?

As detailed above, my main concerns are related to the reported computational data.

*** Suggested improvements: Please list additional experiments or data that could help strengthening the work in a revision.***

See section data and methodology.

There are some errors/typos in the text. A careful read through for checking grammar errors is suggested.

References: Does this manuscript reference previous literature appropriately? YES

*** Clarity and context: Is the abstract clear, accessible? Are abstract, introduction and conclusions appropriate?***

In general, yes with the exception of what noted above.

***Inflammatory material: Does the manuscript contain any language that is inappropriate or potentially libelous? ***

NO, it does not

*** Springer Nature is committed to diversity, equity and inclusion; please raise any concerns that may in your view have an impact on this commitment.***

Does not apply

*** Please indicate any particular part of the manuscript, data, or analyses that you feel is outside the scope of your expertise, or that you were unable to assess fully.*** My expertise is mainly in the computational field. I have no experience with KPFM.

Reviewer #3 (Remarks to the Author):

The paper describes an procedural analysis on how and why a recently described all organic heterojunction, PHI-PTI, generates its record high efficiencies in overall water splitting and CO₂ reduction. Contrary to previous work which generates its solid-state heterojunction photochemical material by simultaneous cocrystallization and salt melt etching, the present approach is based on independent crystallization and later electrostatic co assembly, which gives a larger structural size enabling to follow photogenerated charges, e.g. by surface potential measurements. With a combination of data, the presence of a S-heterojunction as well as the relative role in oxidation and reduction processes was determined.

This makes the paper interesting and worth the citation by a very large, very competitive community. This also brings me to my very important point of criticism dealing with correct citations within the ultracompetitive asian market of artificial photosynthesis.

To make a long analysis short: the manuscript essentially cites one of the two schools, even overscribing priority of the other school by citing later secondary papers of the first school. Being not involved in this specific fight for leadership, I find it nevertheless non acceptable. I seriously and kindly ask the authors to consider the rules of fairness and respect.

One specific example: The first publication on PHI/PTI heterojunctions are not refs 23-25. but work by the Fuzhou-team from 2018.(GG Zhang et al. DOI10.1002/anie.201804702, Ionothermal Synthesis of

Triazine-Heptazine-Based Copolymers with Apparent Quantum Yields of 60% at 420nm for Solar Hydrogen Production from "Sea Water", as well as DOI10.1002/anie.201706870). The same group later expanded this system to photochemical CO₂ reduction, also with record high conversion values (DOI10.1002/anie.201811938). The heterojunction concept as well as driving both artificial photosynthesis reaction on the reduction side with record high efficiencies was thereby known before.

In digital age, such things are easy to find out, and I could go on with a localization of where most citations go to and what else was ignored for that, and I would do in a second round if the authors do not get now what principles in scientific publishing are about.

It is to my opinion indeed a very good, careful paper, and unbalanced recitation of the market can only ruin that impression.

Response to reviewers

We thank the editor and referees for their time and valuable comments on our manuscript. We have studied the reviewer's comments carefully and have made conscientious revisions. In the following, we provide a point-by-point response to their comments.

Reviewer #1 (Remarks to the Author):

This manuscript demonstrates the so-called homojunction between two allotropes of carbon nitride and investigated the charge transfer behavior between the two components. The study of the charge carrier transfer at the interface is of significant importance to deeply understanding the photocatalytic mechanism. The main assumption of this submission is attractive, but more solid characterizations are still lacked.

Response: Many thanks to the reviewer for recognizing our work and providing specific and valuable guidance for us to further revise the manuscript. According to the reviewer's suggestion, we supplemented the in-situ XPS, HRTEM, XPS, element analysis measurement (Data provided by ICP-OES and O, N, H analyzers), and photoreduction of CO₂ experiment. The specific analysis and discussion were carried out in the point-by-point response. We believe that we have well answered the reviewer's question.

1. The author assume the formation of homojunction between TCN and HCN. However, there are no such characterizations (such as TEM) to provide direct evidence. More detail information at the interface is highly desired to support the main assumption of the manuscript. It is widely considered that the weak electrostatic interaction between TCN and HCN only generate nanocomposites rather than junctions.

Response: We have noted this and supplemented the supporting information with HRTEM images of the homojunction (Supplementary Fig. 6c). The HRTEM images were captured at the junction of the nanotubes and nanosheets and clearly show the presence of both TCN and HCN crystalline phases, implying that there are indeed two crystalline phases in TH1:4. The reviewer is right that generally weak electrostatic interactions form nanocomplexes, but there are some syntheses of homojunction that do follow electrostatic self-assembly synthesis¹.

In addition, the homojunction is characterized by a high lattice match and fast electron transfer at the interface. In the calculation part, the corresponding structural models were matched based on the XRD patterns of the synthesized materials, and the lattices of TCN and HCN were so well matched that the constructed lattice mismatch ratio of the homojunction model is only 0.4%. More importantly, TH1:4 demonstrated faster carrier migration than other samples in electrochemical tests, achieving better carrier separation (Supplementary Fig. 24). This evidence is sufficient to show that homogeneous junctions are formed between TCN and HCN.

2. The author demonstrates high crystallinity of TCN and HCN. But only lattice fringes that are related to the interplanar spacings were recorded by TEM. The lattice fringes related to the in-plane packing modes (100) are instead solid evidence to prove the high crystallinity of the carbon nitride materials. Also, the absence of diffraction peak at 8° of HCN in Figure 1h illustrates that HCN is disorder in the in-plane packing modes of the heptazine units.

Response: We have noted this. The reviewer's mentioned that "lattice streaks associated with the in-plane stacking pattern (100) is strong evidence of the high crystallinity of the carbon nitride material" and the "diffraction peak at 8° " are

for the poly heptazine imide (PHI) material. The PHI (a broadly defined crystalline carbon nitride, actually a derivative of crystalline carbon nitride) material reported by Bettina V. Lotsch *et al.* does have a significant characteristic peak at 8° ^{2,3}. However, after XRD pattern comparison, the HCN in this paper is not a PHI material, but a crystallized heptazine based g-C₃N₄. (The relationship between g-C₃N₄ and PTI, PHI is discussed in detail in a later section)

For g-C₃N₄, the characteristic peak around 27° (corresponding to (002) crystal plane) is the dominant peak, which means that the lattice striations related to interplanar spacing are one of the strong evidence to prove high crystallinity.

In addition, the peaks of HCN in this work appear right-shifted and the peak shape becomes sharp around 27° compared to that of the BCN, which is consistent with the literature reports^{4,5}, and the transmission electron microscopy reveals clear lattice stripes, which is sufficient evidence for the high crystallinity of HCN. In addition, the characteristic peak at 13.1° related to the in-plane stacking mode (100) is very weak, which is caused by the curling folds of the ultrathin nanosheets⁶.

In fact, the broad sense of carbon nitride material contains g-C₃N₄, PTI, PHI, etc. At present, many people in the field confuse triazine-based g-C₃N₄, heptazine-based g-C₃N₄, PTI, PHI, with each other, and in fact, they are completely different substances. Strictly speaking, we often refer to carbon nitride materials as g-C₃N₄, but since this kind of material was initially predicted to exist through a computational simulation, the corresponding crystal structure model was simultaneously assumed for the perfect crystallization of this material and adopted till now. However, in fact, after more than a century of development, this material is still not perfectly crystalline when synthesized experimentally⁷. In

order to improve the crystallinity of carbon nitride materials, researchers have tried various methods such as high-temperature and high-pressure methods, ultrasonic methods, and molten salt methods⁸. Among them, the high crystallinity of materials such as PTI and PHI synthesized by molten salt method has attracted much attention. Therefore PTI and PHI were called crystalline carbon nitride in the early days. Strictly speaking, PTI and PHI are derivatives from the process of synthesizing crystalline carbon nitride, and that are not identical to g-C₃N₄.

3.The chemical compositions of so-called TCN and HCN are still not very clear. Are there any residues of the metal salts? Previous studies by Lotsch and Antonietti claimed that Li⁺ and Cl⁻ ions are presented in PTI/Li⁺Cl⁻ (here TCN) and K⁺ always incorporated in K-PHI (here HCN). Please clarify the difference between TCN, HCN and PTI/Li⁺Cl⁻ K-PHI.

Response: TCN and HCN samples had residues of metal salts. As shown in Fig. R1 a, the characteristic peaks of Cl and K were detected in the XPS survey spectra of TCN and HCN, respectively. To further measure the metal element content accurately, we used ICP measurement for HCN and obtained the elemental content of Li and K as 1.81% and 0.934%, respectively (Fig. R1b). Here we need to point out that trace metal salt residues are unavoidable in the preparation of this material, but do not affect the structure and properties of this material⁹.

As noted by the reviewer, previous studies by Lotsch and Antonietti claimed that Li⁺ and Cl⁻ ions are presented in PTI/Li⁺Cl⁻, in this work, after comparison with the XRD pattern of PTI, the TCN in this work is PTI, while HCN is not K-PHI.

Fig. R1 a XPS survey spectra of BCN, TCN, and HCN samples. b Elemental K and Li content in HCN samples.

4. Connected the above question, the acid treatment of HCN would lead to ion exchange (K^+ was replaced by proton). Elemental analysis of the samples before and after acid treatment are involved.

Response: The content of K and Li was obtained from ICP-OES (PE Avio 200) and the content of H was obtained from O, N and H co-meter (LECO ONH836). As shown in Fig. R2, after acid treatment, the content of metallic elements decreases significantly, indicating that acid washing removes some of the metal salts. In fact, our group has studied in detail the effect of acid treatment to eliminate K ions on the photocatalytic activity of crystallized carbon nitride in

our previous work¹⁰. Acid treatment undoubtedly depletes some of the K ions, and the removal of K ions does increase the photocatalytic activity of crystalline carbon nitride, and the highest activity is observed at a hydrochloric acid concentration of 0.1 mol L⁻¹.

However, unexpectedly, the content of H element did not increase after acid treatment as expected, which is because K and Li ions exist in the material in the form of intercalation, and the H ions obtained by replacement cannot be intercalated between the layers and will be removed with centrifugation. On the other hand, the terminal amino group in the carbon nitride material will be removed by hydrochloric acid, so that the final result shows that the content of the H element also becomes less. Although the acid treatment resulted in a small change in the concentration of H ions in the carbon nitride, it did not damage the structure and properties of the material¹¹.

Fig. R2 Comparison of elemental content in HCN samples (before acid washing) and HHCN (after acid washing) samples.

5. Here they author show the nanosheet structure of HCN, which is quite different from the nano rode/fiber morphology of the K-PHI in the previous literature. Please add more explanation about the morphology.

Response: We have noted this, and according to XRD analysis, HCN does not show the characteristic peak of K-PHI at 8° ; therefore, we cannot attribute HCN to K-PHI, and the HCN synthesized in this work is crystallized heptazine-based crystalline carbon nitride (g-C₃N₄). The morphology of heptazine-based g-C₃N₄ is in the form of nanosheets¹².

As the reviewer stated that different morphologies do occur during the preparation of crystalline carbon nitride. According to our previous research experience, the precursor morphology, calcination atmosphere, calcination temperature, and time all affect the degree of crystallization and morphology of crystalline carbon nitride¹³.

Generally speaking, the synthesis of heptazine-based crystalline carbon nitride requires a two-step process. First, the precursor is prefired to synthesize melon, and then further condensed in molten salt to form crystallized carbon nitride^{14,15}. We have carefully studied the literature that reported the nanorod morphology and found that their calcination atmosphere is different from ours¹⁶. In the literature, the melon material was recrystallized in air, which lacks an inert atmosphere, whereas in this experiment, it was recrystallized in a nitrogen atmosphere.

Recrystallization in a nitrogen atmosphere not only promotes the kinetic mass transfer process and improves the crystallinity, but also exfoliates the bulk precursors into nanosheets by thermal exfoliation¹⁷. However, if there is a lack of protection by an inert atmosphere, the crystallization process of carbon nitride is

easily disturbed by impurities and the kinetic mass transfer process is slow, leading to the formation of nanorods or nanofibers in the interstices of molten salt.

6. In-situ photo-deposition of Pt nanoparticles from chloroplatinic acid are usually observed in literatures, but here PtO rather than Pt is obtained, please comment on this issue.

Response: The reviewer is right about this point. Theoretically, photoexcited electrons can reduce Pt⁺⁶ in chloroplatinic acid to Pt nanoparticles. However, the actual detected Pt 4f pattern indicates that the photoreduced Pt species is PtO. There may be two reasons for this. One is that Pt⁺⁶ is not completely reduced due to the limited photoreduction ability, but is only reduced to Pt⁺². The other is the oxidation of the surface of Pt nanoparticles. It is worth noting that this phenomenon is not strange. In fact, some literature also reported that the reduction of Pt⁺⁶ by photodeposition would eventually become PtO instead of Pt nanoparticles^{18,19}.

In this work, the purpose of photoreduction of Pt species is to deposit the reduced Pt species in the conduction band of TCN, thus acting as an electron transfer indicator. PtO is also a product of the reduction and therefore does not affect the conclusion of the article.

7. The characterizations by KPFM is not clearly discussed. Did the author conduct the KPFM direct with the powders of HCN and TCN? As I know, it is usually performing this characterization by using the homogeneous film with flat surface.

Response: The reviewer is right about this point. KPFM cannot directly measure powder samples, which are dispersed in ethanol solution and then dropped onto

the silicon wafer to form a uniform film for testing during our experimental processing. As suggested by the reviewer, the test details of KPFM have been added in the Sample characterization section of the revised manuscript.

In addition, the dynamic transfer of carriers cannot be captured directly if the HCN and TCN samples are measured separately, and with this in mind, we finally measured the TH1:4 sample, choosing the region at the junction of TCN and HCN.

8. In Figure S15, the CO₂ reduction activity dramatically decreased only after few hours' reaction. Is this derived from the weak interaction between the two components?

Response: If there is a weak interaction between TCN and HCN, then the photocatalytic activity will not be significantly enhanced in the first few hours, so we speculate that the decrease in activity is not related to the interaction between the two components. The photoreduction reaction of CO₂ is a multi-step synergistic process. The accumulation of intermediates blocking the active site may be responsible for the decrease in activity (Supplementary Fig. 22).

9. The authors propose that the electron transfer path of TH1:4 under light irradiation is from the conduction band of TCN to the valence band of HCN. And the electrons in TH1:4@PtMn take another path, i.e., from the conduction band of TCN to PtO, then to MnO_x and finally to the valence band of HCN. Does this mean that PtO on TCN and MnO_x on HCN also form good contacts and facilitate electron transfer? The photocatalytic activity of TH1:4@PtMn is worth investigating.

Response: The reviewer is right. Theoretically PtO on TCN and MnO_x on HCN will also form good contacts and promote electron transfer so the photocatalytic

activity of TH1:4@PtMn in principle will exceed that of TH1:4. However, we investigated the photocatalytic activity of TH1:4@PtMn and found that the photocatalytic activity of TH1:4 decreased significantly after loading the double co-catalyst and exhibited a similar photocatalytic activity to that of TCN (Fig. R3a, b). Combined with the available data, the possible reason is that the loading of Pt species leads to the migration of the active site from the conduction band side of HCN to the conduction band side of TCN, thus losing part of the reduction driving force (Fig. R3c). The loading of Pt species on TCN is actually detrimental to interfacial electron transfer. As analyzed in the main text, Pt is involved in two electron transfer processes, and the electron transfer from TCN to Pt dominates, with a large number of electrons concentrated on Pt, thus becoming the main site for CO₂ reduction.

Fig. R3 a Yields of CO and CH₄ of TH1:4@PtMn, and b comparison of CO and CH₄ yields of sample TH1:4@PtMn and TH1:4. c Mechanistic diagram of photocatalytic reduction of CO₂ on TH1:4 and TH:4@PtMn surfaces.

10. The XPS Pt 4f and Mn 2p spectra (in dark, light on and light off) of TCN@Pt, HHCN@Mn, TH1:4@MnPt (MnO_x on TCN and PtO on HCN) is beneficial for further understanding of the electron transfer mechanism. Can the author provide these data?

Response: We have provided the following data as requested. The electron transfer path of the TCN@Pt sample under photoexcitation conditions is depicted in Fig. R4a. It can be seen from the figure that the spectra of Pt 4f is shifted to the left after illumination compared to that of the dark state, which means that the photoelectrons excited by light on TCN are transferred to Pt (Fig. R4b). For HCN@Mn, the spectrum of Mn 2p is shifted to the right after illumination compared with that of the dark state, which means that Mn loses electrons under photoexcitation conditions and electrons may be transferred from Mn to HCN, in other words, holes in the valence band of HCN are transferred to Mn for further oxidation of MnO_x.

Fig. R4c and d depict the electron migration paths of TH1:4@MnPt under photoexcitation conditions. Th1:4@MnPt means that Pt is loaded on the conduction band of HCN and Mn is loaded on the valence band of TCN. The binding energy of Pt 4f becomes smaller in TH1:4@MnPt upon photoexcitation compared to that of the dark state, which indicates that electrons are transferred from HCN to Pt and this process dominates (Fig. R4e). The binding energy of Mn 2p in TH1:4@MnPt also appears to decrease upon photoexcitation compared to that of the dark state, which indicates the transfer of electrons from the conduction band of TCN to Mn and somehow mitigates the carrier recombination due to the role of Mn in receiving electrons (Fig. R4f, g).

Fig. R4 a XPS Pt 4f spectra of TCN@Pt and b the corresponding electron transfer schematics. c XPS Mn 2p spectra of HCN@Mn and d the corresponding electron transfer schematics. e XPS Pt 4f spectra and f Mn 2p spectra of TH1:4@MnPt HCN@Mn and g the corresponding electron transfer schematics.

Reviewer #2 (Remarks to the Author):

Key results: Please summarise what you consider to be the outstanding features of the work.

Understanding how the e- and h+, produced in the bulk of a photocatalyst transfer and eventually react with adsorbed substrates, is very important to rationally design and optimize the photocatalyst.

The authors aim at clarifying the mechanism of electron transfer at the homojunction of crystalline TCN (triazine carbon nitride) and HCN (heptazine carbon nitride) with a combined experimental (TEM, HRTEM, XRD, DRS, VB XPS, UPS, EDS, KPFM, AFM) and computational approach (DFT calculations). Surface bimetallic cocatalysts are used as probes to verify the scheme e- transfer from TCN to HCN. The investigation has a more general goal to define a strategy to probe the electron transfer mechanism, which is relevant to a broad audience.

Response: Many thanks to the reviewers for their appreciation and recognition of this work. This work uses crystalline triazine/heptazine carbon nitride homojunctions as a model system to capture dynamic charge transfer using advanced in situ photoemission spectroscopy and scanning probe microscopy combined with theoretical calculations, and providing a general guide for studying the electron transfer mechanism to design efficient material structures for energy storage.

Validity: Does the manuscript have flaws which should prohibit its publication? If so, please provide details.

A good part of the presented data is solid, and the investigation was well organised. However, in the present form, the reliability of the computational data is not clear, my concerns are reported in the section “Data & methodology” below. The validity of the computed data should be better discussed and possibly verified with further

calculations, as suggested below.

Response: We have taken note of the reviewers' concerns regarding the computed part of the data, and as suggested by the reviewers, the relevant calculations have been double-validated and the computational details have been added to the revised manuscript.

Originality and significance: If the conclusions are not original, please provide relevant references. On a more subjective note, do you feel that the results presented are of immediate interest to many people in your own discipline, and/or to people from several disciplines?

The results are potentially interesting and relevant to a broad audience of people working with photosensitized systems.

Response: We thank the reviewers for recognizing the novelty and importance of this work, which proposes a novel strategy for tracking charge migration that will greatly contribute to the study of charge separation dynamics and has implications for information and energy applications, such as optoelectronics, photovoltaics, and artificial photosynthesis.

Data & methodology: Please comment on the validity of the approach, quality of the data and quality of presentation. Please note that we expect our reviewers to review all data, including any extended data and supplementary information. Is the reporting of data and methodology sufficiently detailed and transparent to enable reproducing the results?

The methodology used in the DFT calculations is not clear, and therefore the criteria of reproducibility is not met.

Response: We have taken care of this and have added methodological details to

the Theoretical details section of the revised manuscript as well as to Supplementary Table. 3 and Supplementary Fig. 13 in the Supporting Information, as suggested by the reviewers.

1) It is not completely clear what systems were simulated (see below). How many atoms were included in the periodic cells? how many ions?

A clear indication of the modeled system(s) comprising also a proper representation of the system simulated is fundamental to enable reproducing the results. You can find many examples of clear representation of the periodic simulation box in the literature, as examples: Phys. Chem. Chem. Phys., 2016, 18, 31175—31183, Fig. 1 for representation of the simulation cell for the junction; J. Phys. Chem. C 2018, 122, 7712–7719, Fig. 1 for the representation of g-C₃N₄ supercell model; nature communications 2020 11:4613, Fig. 3, for box representation.

Response: We have paid attention to this issue. As suggested by the reviewer, we have added and clearly described the simulated model system. The structural model of bulk HCN was obtained from the Material Project database, and the structure model of bulk TCN was obtained from the literature²⁰. As shown in Fig. R5, the constructed homojunction model contains 33 atoms (1 chloride ions, and 3 lithium ions). The specific atomic species and numbers have been added to the Supporting Information in the form of a table (Supplementary Table. 3).

Specific details involving constructing structural model simulated in this work are as follows: The slab models of TCN and HCN were obtained by cleaving (002) crystal planes of bulk TCN and bulk HCN. The monolayer slab model of TCN and the monolayer slab model of HCN are geometrically optimized accompanied by a vacuum layer thickness of 15 Å. The information on the work function is then obtained by property calculations. The crystallized carbon nitride homojunction model was constructed using a monolayer slab of

TCN and a monolayer slab of HCN and the lattice mismatch was within 5% (0.4%). The vacuum layer thickness of the homojunction model is set to 20 Å. The optimized structural parameters for the homojunction model are as follows, $a=b=8.0234$ Å, $c=25.1154$ Å. Light irradiation conditions were simulated by an applied electric field along the z direction.

Fig. R5 a Side view of bulk HCN. b Top view of optimized monolayer HCN with selected crystallographic planes as (002) planes. c Side view of bulk TCN. d Top view of optimized monolayer TCN with selected crystallographic planes as (002) planes. e Side view and f top view of optimized crystalline carbon nitride homojunction.

It is not clear how many systems were modeled: TCN alone, HCN alone, and the

TCN/HCN homojunction? Or only the latter? Are Li^+ and Cl^- included in the simulation box? Please state clearly this information in the method section (a Table might be useful) and represent all of the model systems clearly (see examples above) either in the main text or in the supporting.

Response: We simulated three systems, the monolayer of HCN, the monolayer of TCN, and triazine/heptazine crystalline carbon nitride homojunction. The simulated structures involving TCN contain Cl^- and Li^+ . We have added relevant information to the methodology and Supporting Information as requested by the reviewer.

2) the separation between the periodic structures along the z-axis is set to 15 Å. This value is smaller than that of 20 Å used in other investigations to avoid spurious interactions between periodic image (see e.g. NATURE COMMUNICATIONS | (2020) 11:4613; J. Phys. Chem. C 2018, 122, 7712–7719; Applied Catalysis B: Environmental 2020, 268, 118381). While the value of 15 Å has been used also in other calculations, my concern here is due to the presence of ions, which could make the interaction between layers be stronger (i.e. longer ranged) than for other systems were neutral layers are modeled. I suggest the author to check whether the separation may affect the results and the conclusions, or if there are other evidences that indicate that the use of this separation does not alter the results on similar systems.

Response: We have paid attention to this issue and we have recalculated the work functions of monolayer HCN and TCN with a vacuum layer thickness of 20 Å (Fig. R6, Table R1). The obtained results are not much different from the previous ones, which prove that the distance of 15 Å is enough to resist the effect of van der Waals forces. It is also reported in some literature that the vacuum layer thickness is also set to 15 Å in some PTI (TCN) related calculations²¹, so from both the abovementioned points, the adoption of a 15 Å vacuum layer does

not affect the experimental results and conclusions.

Tabel R1 Relationship between vacuum layer thickness and work function of the sample.

Sample	Work function of 15 Å	20 Å
TCN	6.03 eV	6.07 eV
HCN	4.69 eV	4.62 eV

Fig. R6 a Work function of (002) crystal plane for HCN and b (002) crystal plane for TCN at a vacuum layer thickness of 20 Å.

3) What do the authors mean with “simulated scanning tunneling microscopy (STM) images” (lines 125-127, 573, 577) and what they are used for. Do the authors refer to a STM images calculated as in reference: Scientific Data volume 8, Article number: 57 (2021)? If so, I have the following concern.

If I understand correctly, in the present manuscript no STM experiment is performed. Could the author comment why the simulated STM images are useful in the present investigation if there is no experimental reference? It is not obvious how a simulated STM image can “unveil triazine and heptazine structures as the basic units of TCN

and HCN”: if the structures were used as input for the simulations of the STM spectra, what else could be expected?

Response: The reviewer is right about this point. It is difficult to perform STM experiments because carbon nitride materials are sensitive to high-energy electron beams. Many ambiguous concepts in the field of crystalline carbon nitride often confuse readers when they approach this field, such as the relationship between crystalline carbon nitride and bulk phase carbon nitride, are they the same structure? What is the relationship between triazine-based crystalline carbon nitride and heptazine-based crystalline carbon nitride and PTI and PHI? As you know, there are many geometries of crystalline carbon nitride, and we determined the crystal structure based on XRD diffraction data of the material by comparison. We use STM characterization in this work to explain to the reader more visually what kind of structures we synthesize for triazine-based crystalline carbon nitride and heptazine-based crystalline carbon nitride.

4) Please specify how the work functions were calculated.

Response: Taking the calculation of the work function of TCN as an example, we first optimized the structural model of the bulk TCN, then cleave the (002) crystal plane, followed by adding a vacuum layer, and then geometrically optimized the surface model. After the successful geometric optimization, the property calculation is carried out and the information on the work function of the (002) crystal plane of TCN along the Z-axis can be obtained by analyzing the electrostatic potential of the optimized model.

5) Charge density maps: the analysis of this property has an important role in this investigation to supports the proposed S-scheme mechanism. However, the data

presented are not clear, and also the discussion and conclusions reached with the presented data are not convincing in the present form. In particular, in Sup. Inf. Fig. 9: the pictures (a) and (b) are taken from different points of view and are difficult to compare. Exactly the same orientation should be used to allow the reader to make the comparison and verify the conclusion of the authors. Fig (c) and (d): there are literally infinite possible planes on which the charge density can be represented. Depending on which plane is taken completely opposite conclusion can be (in principle) drawn. Why the authors draw their conclusions only one particular plane? How the conclusions drawn on a particular plane can be considered general?

Response: We have already paid attention to this point. Supplementary Fig. 9a and b (Supplementary Fig. 14a and b in revised Supporting Information) were chosen to be placed at different angles because the difference is obvious to the reader. As suggested by the reviewer we have rearranged the images, taking exactly the same orientation and setting both isovalues to $0.003 e/\text{\AA}^3$. Combined with the Mulliken charge analysis (Fig. 5b in manuscript), it can be seen that under dark conditions, electron depletion occurs on the surface of HCN, whereas electron enrichment emerges on the surface of TCN, revealing electron transfer from HCN to TCN. However, under light conditions, the electron transfer path is completely reversed.

The reviewer is right that different planes can indeed lead to completely different conclusions, but our object of study here is a homojunction composed of TCN and HCN, and the interfacial charge density at the homojunction is the focus of our attention. The (002) crystal plane of TCN and HCN is chosen because the (002) crystal plane of TCN and the (002) crystal plane of HCN are in contact in the constructed homojunction, and the charge transfer at the interface often occurs in this region. Therefore, the contact interface we choose here is universal for the study of interfacial charge transfer in homojunction.

In addition, we would like to clarify the angle of the two-dimensional charge density section. The purpose of the section is to see the interlayer electron transfer clearly, so one criterion for selecting the section is to cut to the most atoms as much as possible, which will give a more objective and comprehensive view of the interlayer electron transfer. If we twist the section by 90°, we will lose a lot of electron transfer information of the C-N bond. Therefore, we have chosen to cut in the horizontal direction here.

6) Mulliken charges are known to have several drawbacks, and other schemes for calculating the charges partitioning among atoms have been developed. Considering that the authors are proposing a strategy of general interest, it could be useful if they can comment on the validity of this aspect of the computational approach they are proposing.

Response: The reviewers are right that there are other schemes for calculating interatomic charge partitioning besides Mulliken charge, a Bouguerian analysis method employed when using atomic orbital basis groups, which is one of the most widely used schemes for calculating atomic charge. Despite being the oldest atomic charge calculation scheme, the Mulliken charge is not obsolete and, like other atomic charge calculation schemes, is constantly being refined. We believe that the Mulliken charge analysis in this work is valid because the plane wave basis set involved is small, which has little effect on the error produced by the Mulliken charge, and because we are analyzing the change in electron density before and after illumination, which is a relative trend. Moreover, the validity of Mulliken charge for charge analysis is currently being trusted by many articles^{22, 23}.

7) Figures

Some Figures are not clear, and improving their quality would be helpful for the

reader:

- Fig 1: the represented molecular structures in insets of subfigure (a) and (d) are not clear. I suggest using more conventional representation in ball and stick. Also, the atom scheme coloring is a bit confusing, it would be easier for the reader to always have gray color for C atoms and blue for Nitrogen, as used in other Figures, such as supporting Fig 9. Same problem in Fig. 5c, it is very difficult to visualize the structure with the representation used here for C and N, and it is not clear why some yellow balls (nitrogen) are not linked to any carbon atoms.. Also, I find confusing the use of two different molecular representations in the same figure (in fig 5a and b is ball and stick with different colors from the vdw representation in fig 5c).

Response: We have made changes to the images in the revised manuscript as suggested by the reviewer. Specifically, all structural models have been uniformly represented by the ball and stick model. The coloring of the atomic scheme has also been set uniformly according to the reviewer's suggestion.

- Fig 2f: it would be easier to analyse if the y axis values would be aligned among the three subfigures. The quality of the figure is low, both in term of resolution and in terms of “collaging” quality (alignment and styles of the graphics).

Response: We have noted this issue and have re-optimized the images in the revised manuscript as suggested by the reviewers. Specifically, we kept the Y-axis values of the three subgraphs the same and removed the background fill from the images, and then standardized the color of the text in the images to black.

- Fig 3e: the writing over the figure is very difficult to read.
- 8) The description of the Figures is in some case not clear:

Response: We have noted this issue and optimized Fig 3e by re-adding the description of the figure.

- Fig. 4h, contains three subfigures, I, II, III. The legend does not clarify what each of them represents.

Response: We have noted the problem and re-added the description of the legend in Fig. 4h. Processes I and II represent the formation of the built-in electric field owing to the difference in the work function, and process III represents the separation of the charges following an S-scheme mechanism induced by the built-in electric field.

- Figs 2b and 5a: what the insets are representing? Why some of the carbon atoms are not represented? Or are they lacking from the model?

Response: The inset shows the geometric model diagram used to calculate the work function. The perspective we put in the original diagram was not consistent, resulting in what appears to be missing of carbon atoms (Top view of the monolayer TCN structure in Fig. 2a and side view of the monolayer HCN structure in Fig. 2b). The correction has been made in the revised manuscript to unify the top view put in.

- Fig. 2a and 5c: what the pink and cyan balls are representing?

Response: The pink and cyan balls represent Li^+ and Cl^- . This part has been supplemented in the caption.

- 9) Other comments on the text:

- Line 129: “TH1:4, i.e. the ratio of heptazine/triazine is 1:4” is this phrase correct? I find confusing that the order of the initials in the acronym is opposite to the ratio of the components to which the initials refers. It would be clearer as HT1:4 if the ratio H/T is 1:4.

Response: We have noted the problem and the reviewer is right, this is the place where we made a mistake in expression. TH1:4, i.e. the ratio of triazine/heptazine is 1:4, and we have corrected the relevant statement in the revised manuscript (Page 6, line 127).

- Line 223-224: here the authors comment on the electrostatic potential, but I cannot find where the results on this property are reported.

Response: We have already noticed this, in fact, the electrostatic potential mentioned here is the value of the y-axis during the calculation of the work function. When the homogeneous junction calculates the work function, the value of the TCN and HCN components corresponding to the y-axis is their electrostatic potential.

Appropriate use of statistics and treatment of uncertainties: All error bars should be defined in the corresponding figure legends; please comment if that’s not the case. Please include in your report a specific comment on the appropriateness of any statistical tests, and the accuracy of the description of any error bars and probability values.

No statistical analysis is reported, the possible errors are not commented.

Response: According to previous experimental tests, the data error is generally within 5%.

Conclusions: Do you find that the conclusions and data interpretation are robust, valid and reliable?

As detailed above, my main concerns are related to the reported computational data.

Response: We have read the reviewers' concerns carefully and have responded to each of them.

*** Suggested improvements: Please list additional experiments or data that could help strengthening the work in a revision.***

See section data and methodology.

Response: As suggested by the reviewers, we have added relevant material not only in the section on section data and methodology but also in the Supporting Information of the revised manuscript

There are some errors/typos in the text. A careful read through for checking grammar errors is suggested.

Response: As suggested by the reviewer, we have read the manuscript carefully and we have noted the grammar errors in the manuscript and have made corrections accordingly.

References: Does this manuscript reference previous literature appropriately?

YES

Response: Yes, this manuscript is appropriately referenced to previous literature.

*** Clarity and context: Is the abstract clear, accessible? Are abstract, introduction

and conclusions appropriate?***

In general, yes with the exception of what noted above.

Response: We have carefully answered all the questions pointed out by the reviewers and have made the corresponding changes in the revised manuscript.

***Inflammatory material: Does the manuscript contain any language that is inappropriate or potentially libelous? ***

NO, it does not

Response: Yes, our manuscripts do not contain inappropriate or defamatory language.

*** Springer Nature is committed to diversity, equity and inclusion; please raise any concerns that may in your view have an impact on this commitment.***

Does not apply

Response: Yes, there are no issues that affect diversity, equity and inclusion in our work.

*** Please indicate any particular part of the manuscript, data, or analyses that you feel is outside the scope of your expertise, or that you were unable to assess fully.***

My expertise is mainly in the computational field. I have no experience with KPFM.

Response: In this work, KPFM was tested on the surface of thin films. The films were prepared as follows: powder samples were dispersed in an ethanol solution and then dropped on the silicon wafer to form a uniform film. The junction region of TCN and HCN was selected with the help of SEM and then the data of surface potential was obtained accompanied by scanning with the probe of

KPFM. The test details of KPFM have been added in the Sample characterization section of the revised manuscript.

Reviewer #3 (Remarks to the Author):

The paper describes an procedural analysis on how and why a recently described all organic heterojunction, PHI-PTI, generates its record high efficiencies in overall water splitting and CO₂ reduction. Contrary to previous work which generates its solid-state heterojunction photochemical material by simultaneous cocrystallization and salt melt etching, the present approach is based on independent crystallization and later electrostatic co assembly, which gives a larger structural size enabling to follow photogenerated charges, e.g. by surface potential measurements. With a combination of data, the presence of a S-heterojunction as well as the relative role in oxidation and reduction processes was determined.

This makes the paper interesting and worth the citation by a very large, very competitive community.

This also brings me to my very important point of criticism dealing with correct citations within the ultracompetitive asian market of artificial photosynthesis.

To make a long analysis short: the manuscript essentially cites one of the two schools, even overscribing priority of the other school by citing later secondary papers of the first school. Being not involved in this specific fight for leadership, I find it nevertheless non acceptable. I seriously and kindly ask the authors to consider the rules of fairness and respect.

One specific example: The first publication on PHI/PTI heterojunctions are not refs 23-25. but work by the Fuzhou-team from 2018.(GG Zhang et al. DOI10.1002/anie.201804702, Ionothermal Synthesis of Triazine-Heptazine-Based Copolymers with Apparent Quantum Yields of 60% at 420nm for Solar Hydrogen Production from "Sea Water", as well as DOI10.1002/anie.201706870). The same group later expanded this system to photochemical CO₂ reduction, also with record high conversion values (DOI10.1002/anie.201811938). The heterojunction concept as well as driving both artificial photosynthesis reaction on the reduction side with record high efficiencies was thereby known before.

In digital age, such things are easy to find out, and I could go on with a localization of where most citations go to and what else was ignored for that, and I would do in a second round if the authors do not get now what principles in scientific publishing are about.

It is to my opinion indeed a very good, careful paper, and unbalanced recitation of the market can only ruin that impression.

Response: It is a great honor to have this work rated so highly by the reviewer, which makes the joint efforts of all the authors over the past two years meaningful.

Regarding the issue of correct citation raised by the reviewers, we must admit that the early studies on PHI/PTI heterojunctions were indeed overlooked by us and we are very grateful to the reviewers for the reminder. We have carefully studied the literature mentioned by the reviewers²⁴⁻²⁷. This literature focuses on the modulation of the grain boundary properties of crystalline carbon nitride for exciton dissociation and the construction of intramolecular triazine-heptazine frameworks. These works are indispensable references for research in the field of crystalline carbon nitride.

The reviewer is right that fair and unbiased citation is a respect to all original researchers. We have added related literature to the revised manuscript (Ref 23-26).

References

- 1 Huang, H., Xiao, K., Du, X. & Zhang, Y. Vertically aligned nanosheets-array-like BiOI homojunction: Three-in-one promoting photocatalytic oxidation and reduction abilities. *ACS Sustain. Chem. Eng.* **5**, 5253-5264 (2017).
- 2 Schlomberg, H. *et al.* Structural insights into poly(heptazine imides): A

- light-storing carbon nitride material for dark photocatalysis. *Chem. Mater.* **31**, 7478-7486 (2019).
- 3 Savateev, A. *et al.* Potassium poly(heptazine imide): Transition metal-free solid-State triplet sensitizer in cascade energy transfer and [3+2]-cycloadditions. *Angew. Chem. Int. Ed.* **59**, 15061-15068 (2020).
 - 4 Li, Y., Li, B., Zhang, D., Cheng, L. & Xiang, Q. Crystalline carbon nitride supported copper single atoms for photocatalytic CO₂ reduction with nearly 100% CO selectivity. *ACS Nano* **14**, 10552-10561 (2020).
 - 5 Lin, L. H. Ren, W. Asiri, AM. Wang, C. Zhang, J. & Wang, X C *et al.* *Appl. Catal. B Environ.* **231**, 234-241(2018).
 - 6 Xiao, Y. *et al.* Molecule self-assembly synthesis of porous few-layer carbon nitride for highly efficient photoredox catalysis. *J. Am. Chem. Soc.* **141**, 2508-2515 (2019).
 - 7 Kroke, E. & Schwarz, M. Novel group 14 nitrides. *Coord. Chem. Rev.* **248**, 493-532 (2004).
 - 8 Lin, L. H., Yu, Z. Y. & Wang, X. C. Crystalline carbon nitride semiconductors for photocatalytic water splitting. *Angew. Chem. Int. Ed.* **58**, 6164-6175, (2019)
 - 9 Liang, X. *et al.* The directional crystallization process of poly (triazine imide) single crystals in molten salts. *Angew. Chem. Int. Ed.* e202216434, (2023).
 - 10 Li, Y., Zhang, D., Feng, X. & Xiang, Q. Enhanced photocatalytic hydrogen production activity of highly crystalline carbon nitride synthesized by hydrochloric acid treatment. *Chin. J. Catal.* **41**, 21-30 (2020).
 - 11 Kim, H., Jang, D., Choi, S., Kim, J. & Park, S. Acid-activated carbon nitrides as photocatalysts for degrading organic pollutants under visible light. *Chemosphere* **273**, 129731 (2021).
 - 12 Zhang, X. D. *et al.* Enhanced photoresponsive ultrathin graphitic-phase C₃N₄ nanosheets for bioimaging. *J. Am. Chem. Soc.* **135**, 18-21 (2013).
 - 13 Li, Y., Zhang, D., Fan, J. & Xiang, Q. Highly crystalline carbon nitride hollow

- spheres with enhanced photocatalytic performance. *Chin. J. Catal.* **42**, 627-636 (2021).
- 14 Zhou, M. et al. Poly(heptazine imide) with enlarged interlayers spacing for efficient photocatalytic NO decomposition. *Appl. Catal. B Environ.* **317**, 121719 (2022).
- 15 Burrow, J. N. et al. Calcium poly(heptazine imide): A covalent heptazine framework for selective CO₂ adsorption. *ACS Nano* **16**, 5393-5403 (2022).
- 16 Zhai, B. et al. A crystalline carbon nitride based near-infrared active photocatalyst. *Adv. Funct. Mater.* **32**, 2207375 (2022).
- 17 Ou, H. et al. Tri-s-triazine-based crystalline carbon nitride nanosheets for an improved hydrogen evolution. *Adv. Mater.* **29**, 1700008 (2017).
- 18 Yang, J. X. et al. PtO nanodots promoting Ti₃C₂ MXene in-situ converted Ti₃C₂/TiO₂ composites for photocatalytic hydrogen production. *Chem. Eng. J.* **420**, 129695 (2021).
- 19 Jiang, J., Yu, J. & Cao, S. Au/PtO nanoparticle-modified g-C₃N₄ for plasmon-enhanced photocatalytic hydrogen evolution under visible light. *J. Colloid Interface Sci.* **461**, 56-63 (2016).
- 20 Wirnhier, E. et al. Poly(triazine imide) with intercalation of lithium and chloride ions (C₃N₃)(2)(NH_xLi_{1-x})(₃)center dot LiCl : A crystalline 2D carbon nitride network. *Chem. Eur. J.* **17**, 3213-3221 (2011).
- 21 Shen, S. et al. Boosting photocatalytic hydrogen production by creating isotype heterojunctions and single-atom active sites in highly-crystallized carbon nitride. *Sci. Bull.* **67**, 520-528 (2022).
- 22 Liu, Y. et al. Silver nanoparticle enhanced metal-organic matrix with interface-engineering for efficient photocatalytic hydrogen evolution. *Nat. Commun.* **14**, 541 (2023).
- 23 Zhang, Y. et al. Single-atom Cu anchored catalysts for photocatalytic renewable H₂ production with a quantum efficiency of 56%. *Nat. Commun.* **13**, 58 (2022).

- 24 Zhang, G. et al. Ionothermal synthesis of triazine–heptazine-based copolymers with apparent quantum yields of 60 % at 420 nm for solar hydrogen production from “Sea Water”. *Angew. Chem. Int. Ed.* **57**, 9372-9376 (2018).
- 25 Lin, L. H. et al. Molecular-level insights on the reactive facet of carbon nitride single crystals photocatalysing overall water splitting. *Nat. Catal.* **3**, 649-655 (2020).
- 26 Zhang, G. et al. Optimizing optical absorption, exciton dissociation, and charge transfer of a polymeric carbon nitride with ultrahigh solar hydrogen production activity. *Angew. Chem. Int. Ed.* **56**, 13445-13449 (2017).
- 27 Zhang, G. et al. Tailoring the grain boundary chemistry of polymeric carbon nitride for enhanced solar hydrogen production and CO₂ reduction. *Angew. Chem. Int. Ed.* **58**, 3433-3437 (2019).

REVIEWER COMMENTS

Reviewer #1 (Remarks to the Author):

The authors provide more information of the material and the revised version of manuscript was indeed improved. However, I still would like to learn more details at the interface of the HCN and TCN. How the heptazine-based units were connected with the triazine-based units with different packing modes. Is it connected with semi-coherent interface? The energy barrier at the interface for charge diffusion is also desired. More solid characterizations such as HR-TEM images at the interface are of significant importance to guide the development of carbon nitride family.

Reviewer #2 (Remarks to the Author):

The authors have fully addressed the remarks made by me and by the other reviewers, and the manuscript is now ready for publication.

Reviewer #3 (Remarks to the Author):

The authors have addressed the raised questions and concerns of the first set of reports accordingly, and the manuscript has remarkably improved. According to the good habits of scientific refereeing, I now have to support publication without change.

Response to reviewers

We thank the editor and referees for their time and valuable comments on our manuscript. We have studied the reviewer's comments carefully and have made conscientious revisions. In the following, we provide a point-by-point response to their comments.

Reviewer #1 (Remarks to the Author):

The authors provide more information of the material and the revised version of manuscript was indeed improved. However, I still would like to learn more details at the interface of the HCN and TCN. How the heptazine-based units were connected with the triazine-based units with different packing modes. Is it connected with semi-coherent interface? The energy barrier at the interface for charge diffusion is also desired. More solid characterizations such as HR-TEM images at the interface are of significant importance to guide the development of carbon nitride family.

Response: (1) We have noticed this issue and the reviewer is right about it. Due to the differences in the basic structural units between TCN and HCN, the homojunction formed by HCN and TCN has a certain lattice mismatch rate (0.4%), thus the interface between HCN and TCN is a semi-coherent interface¹.

(2) More detailed information about the interface between HCN and TCN is further elucidated by combining TEM characterisation and structural model analysis from the theoretical calculations. Fig 1a-c clearly demonstrate the structural model of a homojunction composed of TCN and HCN, with an interlayer spacing of 3.3 Å, indicating that the interlayer interaction force belongs to the van der Waals force. To further reveal the information at the interface, we depicted a schematic diagram of the atomic arrangement at the interface by combining XRD analysis and TEM images (Fig.1d and e). The exposed crystal planes of TCN and HCN are (002) crystal planes stacked between layers, and TCN

and HCN are connected at the interface by van der Waals force, not by chemical bonds. It is worth noting that the van der Waals force only connects TCN and HCN, and stabilizes the crystalline carbon nitride homojunction. It is not a driving force for interlayer electron transfer.

The energy barrier of electron transport at the interface between layers will be explained detailly in (3). In addition, a detailed schematic of the atomic arrangement has been added to Supplementary Fig. 5d and e in the supporting information, while the atomic arrangement at the interface has been elucidated in the revised manuscript (page 6, line131-133). And as suggested by the reviewer, we have also added TEM characterisation (Fig. 2).

Fig. 1 a, b Side view and c top view of optimized triazine/heptazine crystalline carbon nitride homojunction. d HRTEM images of TH1:4 and e schematic diagram of atomic arrangement at interface.

(3) Regarding the issue of interface energy barrier mentioned by the reviewer, we will answer this question jointly from both theoretical calculations and

experimental characterization. Firstly, it should be pointed out that the driving force for electron transfer at the interface comes from the potential difference between TCN and HCN. As shown in Fig 3a and 3b, the electrostatic potentials between TCN and HCN are -31.27 and -31.88 eV, respectively. Therefore, the energy barrier for interface electron transfer is 0.61 eV, which is the driving force driving electron transfer from HCN to TCN in a dark state environment. This is consistent with experimental characterization.

As analyzed in our manuscript, there is a significant potential difference between TCN and HCN, and the change in interface potential before and after illumination indicates electron transfer. In addition, we also supplemented the Mott-Schottky characterization. According to the slope comparison, the electron concentration of TH1:4 is 1.12 times higher than that of TH1:1², which shows that the electron transfer of TH1:4 is less hindered than TH1:1. Proper phase ratio of triazine and heptazine effectively improves the carrier separation and transmission performance.

Fig. 2 a The work function of the crystallized carbon nitride homojunction, where the inset shows the optimized model of the homojunction. **b** Electrostatic potential difference between layers of crystalline carbon nitride homojunction. **c, d** The distribution of potential difference at the interface of TH1:4 in a dark environment. **e, f** The distribution of potential difference at the interface of TH1:4 under illumination. **g** Schematic of the S-scheme migration of photogenerated carriers induced by the electric field at the TH1:4 interface.

Fig. 3 Mott-Schottky plots of TH 1:4 and TH1:1 collected at 600 Hz.

Reviewer #2 (Remarks to the Author):

The authors have fully addressed the remarks made by me and by the other reviewers, and the manuscript is now ready for publication.

Response: Thank the reviewers for their recognition and appreciation of this work.

Reviewer #3 (Remarks to the Author):

The authors have addressed the raised questions and concerns of the first set of reports accordingly, and the manuscript has remarkably improved. According to the good habits of scientific refereeing, I now have to support publication without change.

Response: Thank the reviewers for their recognition and appreciation of this work.

Reference

- 1 Sun, X. H. *et al.* Dislocation-induced stop-and-go kinetics of interfacial transformations. *Nature*. **607**, 708-+ (2022).
- 2 Wang, H. *et al.* Boosting hot-electron generation: Exciton dissociation at the order disorder interfaces in polymeric photocatalysts. *J. Am. Chem. Soc.* **139**, 2468-2473 (2017).

REVIEWERS' COMMENTS

Reviewer #1 (Remarks to the Author):

The author have fully addressed the concerns from the reviewers, and now it is ready for publication.

REVIEWERS' COMMENTS

Reviewer #1 (Remarks to the Author):

The author have fully addressed the concerns from the reviewers, and now it is ready for publication.

Response: Thank the reviewers for their recognition and appreciation of this work.

Reference

- 1 Sun, X. H. *et al.* Dislocation-induced stop-and-go kinetics of interfacial transformations. *Nature*. **607**, 708-+ (2022).
- 2 Wang, H. *et al.* Boosting hot-electron generation: Exciton dissociation at the order disorder interfaces in polymeric photocatalysts. *J. Am. Chem. Soc.* **139**, 2468-2473 (2017).